# NORM-BOUNDED LOW-RANK ADAPTATION

## ABSTRACT

In this work, we propose norm-bounded low-rank adaptation (NB-LoRA) for parameter-efficient fine tuning. NB-LoRA is a novel parameterization of low-rank weight adaptations that admits explicit bounds on each singular value of the adaptation matrix, which can thereby satisfy any prescribed unitarily invariant norm bound, including the Schatten norms (e.g., nuclear, Frobenius, spectral norm). The proposed parameterization is unconstrained, smooth, and complete, i.e. it covers all matrices satisfying the prescribed rank and singular-value bounds. Natural language generation experiments show that NB-LoRA matches or surpasses performance of competing LoRA methods, while exhibiting stronger hyper-parameter robustness. Vision fine-tuning experiments show that NB-LoRA can avoid model catastrophic forgetting with minor cost on adaptation performance, and compared to existing approaches it is substantially more robust to a hyper-parameters such as including learning rate, adaptation rank and number of training epochs.

## 1 INTRODUCTION

Large pre-trained vision and language models have demonstrated impressive generalization capability across a wide variety of tasks; see, e.g. Achiam et al. (2023); Touvron et al. (2023). When a more specific target task is identified, however, it has been observed that parameter-efficient fine-tuning (PEFT) techniques, e.g. Houlsby et al. (2019); Hu et al. (2022), can improve performance via quick model adaption with low computation and data requirements. The primary goal for an effective PEFT method is to achieve good adaptation performance with high training efficiency, i.e., dramatically fewer trainable parameters and training epochs. Since training efficiency is the target, ideally such a method will be quite robust to hyperparameters. Alongside this primary goal, it is often also desirable to maintain the generalization performance of the original pre-trained model as much as possible, i.e. avoid "catastrophic forgetting" (Qiu et al., 2023; Biderman et al., 2024).

Low-rank adaption (LoRA) (Hu et al., 2022) is a widely applied PEFT method, which parameterizes the update of pretrained weights $W_p \in \mathbb{R}^{m \times n}$ during finetuning as

$$y = (W_p + W)x = \left(W_p + \frac{\alpha}{r} B^\top A\right) x \tag{1}$$

where $A \in \mathbb{R}^{r \times n}$, $B \in \mathbb{R}^{r \times m}$ are the learnable matrices, $\alpha$ is a scaling factor, and $r \ll \min(m, n)$ is the rank budget of weight adaptation $W$. Matrix rank is one way to quantify the "size" of a weight, corresponding the underlying dimensionality of its operation. But matrix norms – such as nuclear, Frobenius, or spectral norms – provide another notion of size, quantifying the magnitude of a matrix's elements and of its operation on vectors.

Recent works show that it is beneficial to control the rank and norm of the weight adaption. Jang et al. (2024); Kim et al. (2025) show that the global minimum of fine-tuning has low rank and small magnitude while spurious local minima (if they exist) have high rank and large magnitude. Moreover, bounding the magnitude of $W$ can enhance training robustness (Bini et al., 2025). In Hu et al. (2025), LoRA training can achieve sub-quadratic time complexity under certain norm-bound conditions.

Motivated by those findings, we propose norm-bounded low-rank adaptation (NB-LoRA), a novel finetuning method that admits explicit bounds on both the rank *and* norm of weight update through matrix reparameterization (see Fig. 1). Our approach can control a family of matrix norms, called Schatten $p$-norms (i.e. $p$-norms of the singular value sequence), which include the nuclear norm, Frobenius norm, and spectral norm as special cases. We summarize our contributions as follows.

Figure 1: Visualization (Left) of the original LoRA (Hu et al., 2022) and (Right) of our proposed method NB-LoRA, where bounded rank and norm are enforced by reparameterization $\mathcal{W}_S$.

- Our parameterization is a smooth map $W = \mathcal{W}_S(\tilde{A}, \tilde{B})$ which takes as argument two free matrix variables of the same size as $A, B$, but the resulting $W$ automatically satisfies user-prescribed bounds on both rank and all individual singular values of $W$, which further allows any Schatten $p$-norm bound on $W$ to be specified.

- Our parameterization is *complete*, i.e., for any $W \in \mathbb{R}^{m \times n}$ satisfying the prescribed bounds on singular values, there exists a (not necessarily unique) $\tilde{A}, \tilde{B}$ such that $W = \mathcal{W}_S(\tilde{A}, \tilde{B})$.

- Theoretical analysis on training dynamics and LLM fine-tuning experiments show that NB-LoRA can improve training stability, overall performance and robustness to learning rates.

- Through ViT fine-tuning and subject-driven image generation tasks, we show that tight bound control can effectively prevent catastrophic forgetting and model overfitting in the low-data regime.

## 2 RELATED WORK

LoRA can be highly sensitive to learning rate (Bini et al., 2024; Biderman et al., 2024), model initialization (Hayou et al., 2024), and it is susceptible to over-training (Qiu et al., 2023). To mitigate these effects, several recent works have proposed regularization techniques for LoRA. For example, Gouk et al. (2021); Chen et al. (2023) propose an approach that preserves the Euclidean weight distances between pre-trained and fine-tuned models. In Liu et al. (2024), DoRA was proposed based on investigation of the vector-wise norm of the adaption matrix, and introduces an adaptive scaling of $W$. Bini et al. (2025) proposed DeLoRA - a PEFT method that decouples the angular learning from adaptation strength. VeRA is another method which learns a scaling vector for LoRA weights (Kopiczko et al., 2024b). Our method also contains a learnable scaling vector, which can be used to explicitly control bounds on each singular value of the weight adaptation.

Another line of LoRA methods are closely related to singular value decomposition (SVD). Meng et al. (2024) proposed a novel SVD-based LoRA initialization, called PiSSA, which can significantly speed up the training of LoRA. Zhang et al. (2023) proposed a dynamical rank allocation scheme, called AdaLoRA, which adaptively update the rank bound in each LoRA layer. In Lingam et al. (2024); Bałazy et al. (2024), the singular vectors of pretrained weights are re-used and a small square matrices are learned during fine-tuning. No explicit control of norm bounds or constraint on singular values were considered in these methods.

## 3 MOTIVATING ANALYSIS OF LoRA

In this section we provide some brief analyses of LoRA that motivate our parameterization.

**Analysis of Training Dynamics.** We first rewrite the LoRA parameterization (1) as the form of

$$W = \hat{B}^\top \hat{A} \tag{2}$$

where $\hat{A} = \sqrt{\alpha/r}A$ and $\hat{B} = \sqrt{\alpha/r}B$. The main purpose of (2) is to give a uniform presentation for analyzing the training dynamics of different low-rank weight parameterizations. Under this

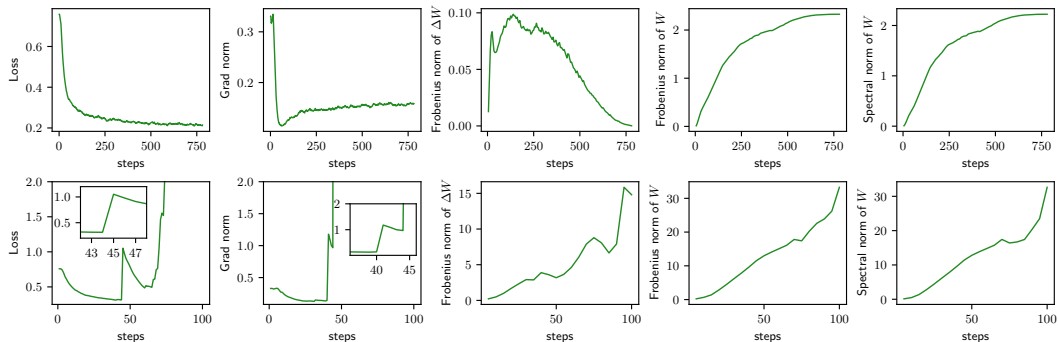

Figure 2: Training dynamics of LoRA ($r = 128$) for LLaMA-2-7B fine-tuning. We report the loss, gradient norm and matrix norms of weight increment $\Delta W$ and weight $W$ under two learning rates: 5e-5 (Top) and 1e-3 (Bottom). Matrix norms are maximized over all LoRA blocks.

representation, the increments on $\hat{A}$, $\hat{B}$ can be approximated by

$$\Delta \hat{A} \approx -\eta \frac{\partial \ell}{\partial \hat{A}} = -\eta \hat{B} D_{xy}, \quad \Delta \hat{B} \approx -\eta \frac{\partial \ell}{\partial \hat{B}} = -\eta \hat{A} D_{xy}^{\top} \tag{3}$$

where $\eta$ is the learning rate, $\ell$ is the loss function, and $D_{xy} = (\partial \ell / \partial y) x^{\top}$ is typically not large at the beginning of fine-tuning. The standard LoRA initialization takes $\hat{B} = 0$ and small random $\hat{A}$, which implies $\Delta \hat{A} = 0$ and $\Delta \hat{B}$ is small and noisy at the beginning of fine-tuning. Then, the weight update $\Delta W$ could be small and uninformative for a large number of training steps, since it depends on $\hat{A}$ and $\hat{B}$ quadratically:

$$\Delta W \approx \hat{B}^{\top} (\Delta \hat{A}) + (\Delta \hat{B})^{\top} \hat{A} = -\eta \big( D_{xy} \hat{A}^{\top} \hat{A} + \hat{B}^{\top} \hat{B} D_{xy} \big). \tag{4}$$

The top row of Figure 2 illustrate this behavior. Increasing the learning rate $\eta$ or scaling factor $\alpha$ can speed up the training but it may cause instability, see the bottom row of Figure 2. This phenomenon has been reported and analyzed in Meng et al. (2024); Zhang et al. (2025), which proposed alternative initializations.

In this work, we provide a novel LoRA reparameterization such that $\|\hat{A}\|_F^2 + \|\hat{B}\|_F^2$ is close to certain constant. This prevents the matrices $\hat{A}$ and $\hat{B}$ becoming simultaneously very small or very large. For example, if $\hat{B}$ is a zero matrix, then by construction $\hat{A}$ is a relatively large matrix, which in turn produces large $\Delta \hat{B}$. As the norm of $\hat{B}$ increases, the norm of $\hat{A}$ will automatically decrease, ensuring that $\hat{B}^{\top} \hat{A}$ remains within certain prescribed norm bound. This coupling behavior can help to improve train stability and robustness.

Although enforcing a bound norm on $W$ may limit the adaptation performance, it could be beneficial for many fine-tuning tasks. For example, when the target dataset $\mathcal{D}_T$ is small, a tight norm bound on $W$ can help to prevent overfitting. Another application is to avoid catastrophic forgetting. After fine-tuning, we can approximate the loss changes on the source dataset $\mathcal{D}_s$ by

$$\ell_{\mathcal{D}_s}(W_p + W) - \ell_{\mathcal{D}_s}(W_p) \approx \frac{1}{M} \sum_{i=1}^{M} \left( \frac{\partial \ell}{\partial y_i^s} \right)^{\top} W x_i^s$$

where $x_i^s, y_i^s$ are the input and output of the pretrained layer, evaluated on $\mathcal{D}_s$. If $\mathcal{D}_S$ is not available, then constraining the norm of $W$ becomes a natural approach. In particular, our parameterization is *complete*, i.e., it covers all weights with the prescribed norm bound. Thus, it can prevent overfitting and catastrophic forgetting with minor cost on adaptation performance.

## 4 NB-LoRA

In this section we present our main contribution: a parameterization of low-rank matrices that admits bounds on each individual singular value, and hence on any unitarily invariant matrix norm.

## 4.1 PRELIMINARIES AND PROBLEM FORMULATION

The problem we are interested in can be formalized as follows:

$$\min \quad \ell(W) \quad \text{s.t.} \quad \text{rank}(W) \leqslant r, \ \|W\|_{S_p} \leqslant \delta \tag{5}$$

where $\ell$ is some training loss and $\|W\|_{S_p} = \left(\sum_{i=1}^r \sigma_i^p\right)^{1/p}$ for $p \in [1, \infty)$ and $\|W\|_{S_\infty} = \sigma_1$, where $\sigma_1 \geqslant \sigma_2 \geqslant \cdots \geqslant \sigma_r \geqslant 0$ are the singular values of $W$. Since Schatten $p$-norm is the vector $p$-norm of the singular value sequence, it is unitarily invariant, i.e., $\|W\|_{S_p} = \|UWV\|_{S_p}$ for any orthogonal matrices $U, V$.

We first define some notation. Since our approach involves comparing singular values of matrices of potentially different ranks and sizes, for convenience we define $\sigma_j(W) = 0$ if $j > \text{rank}(W)$. We now introduce the relation $\preceq_\sigma$.

**Definition 4.1.** Let $X, Y$ be two matrices. We say $X \preceq_\sigma Y$ if $\sigma_j(X) \leqslant \sigma_j(Y), \ \forall j \in \mathbb{N}$.

Note the $\preceq_\sigma$ is reflexive ($X \preceq_\sigma X$) and transitive ($X \preceq_\sigma Y, Y \preceq_\sigma Z \Rightarrow X \preceq_\sigma Z$). But it is not antisymmetric, i.e., $X \preceq_\sigma Y, Y \preceq_\sigma X \nRightarrow X = Y$, e.g., when $X, Y$ are distinct orthogonal matrices. Most importantly for our purposes: if $X \preceq_\sigma Y$, then $\|X\|_{S_p} \leqslant \|Y\|_{S_p}$ for all $p \in [1, \infty]$.

Let $s \in \mathbb{R}_+^r$, where $\mathbb{R}_+ = [0, \infty)$, and $S = \text{diag}(s)$ be the diagonal matrix with $S_{jj} = s_j$. We define the set of matrices whose singular values are bounded by $S$ by

$$\mathbb{W}_S := \{W \in \mathbb{R}^{m \times n} \mid W \preceq_\sigma S\}.$$

Note that for any $W \in \mathbb{W}_S$, we have $\text{rank}(W) \leqslant \text{rank}(S) = r$ and $\|W\|_{S_p} \leqslant \|S\|_{S_p}$.

## 4.2 NB-LoRA PARAMETERIZATION

We now present so-called *direct* parameterization of $\mathbb{W}_S$, a smooth mapping $\mathcal{W}_S$ from free matrix variables to $W$ which maps onto the entire set $\mathbb{W}_S$. Then, we can transform (5) into an unconstrained problem by further parameterizing the positive diagonal matrix $S$ such that $\|S\|_{S_p} = \delta$.

Our parameterization takes $\tilde{A} \in \mathbb{R}^{r \times n}, \tilde{B} \in \mathbb{R}^{r \times m}$ as the free parameters and produces $W$ via

$$W = \mathcal{W}_S(\tilde{A}, \tilde{B}) := 2B^\top S A, \ \text{where} \ \begin{bmatrix} A^\top \\ B^\top \end{bmatrix} = \text{Cayley}\left(\begin{bmatrix} \tilde{A}^\top \\ \tilde{B}^\top \end{bmatrix}\right). \tag{6}$$

Here the Cayley transformation for a tall matrix $\begin{bmatrix} X \\ Y \end{bmatrix}$ with $X \in \mathbb{R}^{r \times r}$ and $Y \in \mathbb{R}^{q \times r}$ is defined by

$$\text{Cayley}\left(\begin{bmatrix} X \\ Y \end{bmatrix}\right) := \begin{bmatrix} (I - Z)(I + Z)^{-1} \\ -2Y(I + Z)^{-1} \end{bmatrix}, \ \text{where} \ Z = X - X^\top + Y^\top Y. \tag{7}$$

Note that $G = \text{Cayley}(F)$ is a semi-orthogonal matrix, i.e., $G^\top G = I$ for any tall matrix $F$ (Trockman & Kolter, 2021), however it is not by itself a complete parameterization for the set of semi-orthogonal matrices, e.g., there does not exist an $F$ such that $\text{Cayley}(F) = -I$. Despite this, we have the following, which is the main theoretical result of the paper.

**Theorem 4.2.** *The NB-LoRA parameterization in (6) is a direct (smooth and complete) parameterization of $\mathbb{W}_S$, i.e. $\mathcal{W}_S$ is differentiable and $\mathcal{W}_S(\mathbb{R}^N) = \mathbb{W}_S$.*

*Remark* 4.3. A special case of the above theorem is $S = I$, which is a complete parameterization of all 1-Lipschitz linear layer, i.e. $f(x) = Wx$ with $\|W\|_{S_\infty} \leqslant 1$, see Proposition 3.3 of Wang & Manchester (2023). One can further extend it to a nonlinear layer with low-rank and norm-bounded Jacobian. Specifically, we take a nonlinear layer of the form $f(x) = 2B^\top D_1 \phi(D_2 Ax)$ where $A, B$ are constructed from (6), $D_1, D_2$ are diagonal matrices satisfying $0 \preceq D_1 D_2 \preceq S$ and $\phi$ is a scalar activation with slope-restricted in $[0, 1]$. Then, we have $\partial f / \partial x \in \mathbb{W}_S$ for all $x \in \mathbb{R}^n$.

**Model initialization.** We take the standard LoRA initialization to NB-LoRA's free parameters: sampling $\tilde{A}$ as a small random matrix and setting $\tilde{B} = 0$. After applying the Cayley transformation, we have $AA^\top = I$ and $B = 0$, yielding a zero initialization for $W$.

**Imposing the Norm Bound on $W$.** From Theorem 4.2, if we construct a complete parameterization for the set of singular bound vector $s \in \mathbb{R}_+^r$ such that $\|s\|_p = \delta$, then the proposed NB-LoRA (6) covers all adaptation matrices $W$ satisfying the prescribed rank and norm bounds. For $p = \infty$, we simply take $s = (\delta, \delta, \ldots, \delta)$. For $p \in [1, \infty)$, one approach is $s = \delta|\tilde{s}|/\|\tilde{s}\|_p$, where $\tilde{s} \in \mathbb{R}^r$ is a free non-zero vector. However, this parameterization is not smooth at $\tilde{s} = 0$. Instead, we use the following parameterization in our experiments:

$$s = \delta\hat{s} := \delta \left[\mathrm{Softmax}\left(\tilde{s}/\sqrt{r}\right)\right]^{1/p}. \tag{8}$$

Technically, it omits some boundary cases with $\|W\|_{S_p} = \delta$ and $\sigma_r(W) = 0$ since softmax has strictly positive outputs. However, since it covers the interior of the feasible set and can approximate the boundary, there is no practical impact on optimization performance. If there is no strict requirement on the norm bound, one can directly learn $s$ via gradient-based or adaptive methods.

**Computational cost of Cayley Transformation.** Due to the low-rank nature ($r$ is often less than 256), computing the inverse of an $r \times r$ matrix is not overly expensive. While matrix inversion is one part of the total training cost, another computationally intensive part is the backward pass for the Cayley transformation (7). We provide an efficient custom backward step in Section C.

## 4.3 Training Dynamics Analysis

Here we return to the motivating analysis from Section 3 and show why NB-LoRA helps resolve the issue of small gradients. We first rewrite NB-LoRA (6) into the uniform representation (2) with $\hat{A} = \sqrt{2}S^{\frac{1}{2}}A$ and $\hat{B} = \sqrt{2}S^{\frac{1}{2}}B$. Then, we have

$$\hat{A}\hat{A}^\top + \hat{B}\hat{B}^\top = 2S^{\frac{1}{2}}(AA^\top + BB^\top)S^{\frac{1}{2}} = 2S. \tag{9}$$

Together with (8) we have that $\hat{A}, \hat{B}$ evolves on a compact manifold of the form

$$\|\hat{A}\|_F^2 + \|\hat{B}\|_F^2 = 2\mathrm{trace}(S) = 2\delta|\hat{s}|_1 := \bar{\gamma}. \tag{10}$$

If nuclear or spectral norm bound is considered, the right hand side of (10) becomes a constant, i.e., $\bar{\gamma} = 2\delta$ or $\bar{\gamma} = 2r\delta$, respectively. For Frobenius norm bound, we have $\bar{\gamma} \in [2\delta, 2\sqrt{r}\delta]$. And $\bar{\gamma}$ is close to $2\sqrt{r}\delta$ as we initialize $\hat{s}_i \approx 1/\sqrt{r}$, i.e., $\tilde{s}$ is initialized as a small random vector. Equation (10) implies that $\hat{A}, \hat{B}$ cannot be both arbitrarily small or large matrices. Thus, $\Delta\hat{A}, \Delta\hat{B}$ and $\Delta W$ have bounded gain w.r.t. $D_{xy}$, allowing stable training for a wider range of learning rates than LoRA.

**PiSSA vs NB-LoRA.** PiSSA (Meng et al., 2024) addresses the small initial gradient issue of LoRA via a residual-type initialization, i.e., $W = \frac{\alpha}{r}(B^\top A - B_0^\top A_0)$ where the initial values of $A, B$ are $A_0, B_0$, respectively. Similar to LoRA, we can cast PiSSA into the form of

$$W = \hat{B}^\top\hat{A} - \hat{B}_0^\top\hat{A}_0 \tag{11}$$

with $\hat{A} = \sqrt{\alpha/r}A$ and $\hat{B} = \sqrt{\alpha/r}B$. Since $\hat{B}_0^\top\hat{A}_0$ is frozen during training, the increments of $\hat{A}, \hat{B}$ follow (3) and (4). The difference is that the residual-type initialization allows one to construct much larger $\hat{A}_0$ and $\hat{B}_0$, see Meng et al. (2024). This can speed up the fine-tuning process, however, its performance might be sensitive to learning rate as $\hat{A}, \hat{B}$ and $W$ are unbounded. Different from PiSSA, NB-LoRA constraints $\hat{A}, \hat{B}$ on a compact manifold defined in (10), which allows for a wide range of learning rates without increasing the norm bounds of $\hat{A}, \hat{B}$ and $W$.

**DeLoRA vs NB-LoRA.** Similarly to our method, DeLoRA (Bini et al., 2025) can also control the Frobenius norm bound of weight adaption based on the following parameterization:

$$W = \frac{\gamma}{2r}B^\top\Xi A - \frac{\gamma_0}{2r}B_0^\top\Xi_0 A_0 \quad \text{with} \quad \Xi = \mathrm{diag}\left(\frac{1}{|a_i|_2|b_i|_2}\right), \tag{12}$$

where $a_i, b_i$ are the $i$th rows of $A, B$ respectively. The scaling factor $\gamma$ and weight parameter $A, B$ are initialized as $\gamma_0$ and $A_0, B_0$, respectively. Similar to PiSSA, we can rewrite DeLoRA into (11 with $\hat{A} = \sqrt{\frac{\gamma}{r}}\mathrm{diag}\left(\frac{1}{|a_i|_2}\right)A$ and $\hat{B} = \sqrt{\frac{\gamma}{r}}\mathrm{diag}\left(\frac{1}{|b_i|_2}\right)B$. From this, we can obtain a similar manifold constraint as NB-LoRA:

$$\|\hat{A}\|_F^2 + \|\hat{B}\|_F^2 = \gamma. \tag{13}$$

When certified Frobenius norm bound of $\delta$ is considered, DeLoRA needs a fixed $\gamma = \gamma_0 = \delta/2$, see Section D. Since the ratio $\bar{\gamma}/\gamma \approx 4\sqrt{r}$, NB-LoRA is more expressive than DeLoRA since it allows for much larger $\hat{A}, \hat{B}$, especially when the rank $r$ is relatively large. This also indicates that DeLoRA needs much larger learning rate than NB-LoRA, see Figure 5. Further discussions on the connections and differences between these two approaches can be found in Section D.

## 5 EXPERIMENTS

Here we evaluate the proposed NB-LoRA approach for natural language generation (NLG), ViT fine-tuning, and image generation tasks. We show that NB-LoRA not only matches or exceeds the performance of LoRA and other related variants but also improves robustness to hyper-parameters.

### 5.1 NATURAL LANGUAGE GENERATION

Our main objectives are as follows: i) NB-LoRA can avoid small initial gradients while still maintain training stability for a wide range of learning rates; ii) Controlling the norm is beneficial for robust performance; iii) Due to the ability of tight bound control, our method can outperform existing approaches with the same certified norm bound.

**NLG Task.** We fine-tuned the LLaMA model family (Touvron et al., 2023) and Mistral-7B-v0.1 (Jiang et al., 2023) on the MetaMathQA dataset (Yu et al., 2023) to evaluate their mathematical problem-solving capability on the GSM8K (Cobbe et al., 2021) and MATH (Hendrycks et al., 2021) test datasets. We also fine-tuned the models on the the CodeFeedback dataset (Zheng et al., 2024) and evaluated for coding proficiency using the HumanEval (Chen et al., 2021) and MBPP (Austin et al., 2021). We adopt the implementation strategy from Taori et al. (2023). We follow the setup in Meng et al. (2024) with default rank $r = 128$ and scaling $\alpha = r$ for LoRa, DoRA and PiSSA, see Section F for more details. The choice of norm bound $\delta$ for NB-LoRA is discussed in Section G.

**Large Initial Gradients and Training Stability.** We conducted experiments on LLaMA-2-7B fine-tuning across a wide range of learning rates from 5e-5 to 1e-3. Figure 3 shows that LoRA and DoRA both suffer from poor performance with small learning rates, due to the small initial gradients. Increasing the learning rate helps up to a point but then training goes unstable. In contrast, NB-LoRA achieves good performance for a wide range of learning rates. PiSSA outperforms NB-LoRA in terms of peak performance on GSM8K, but underperforms on other tasks and is more sensitive to learning rate. In contrast, NB-LoRA achieves good performance for a wide range of learning rates, outperforming all other models on most tasks.

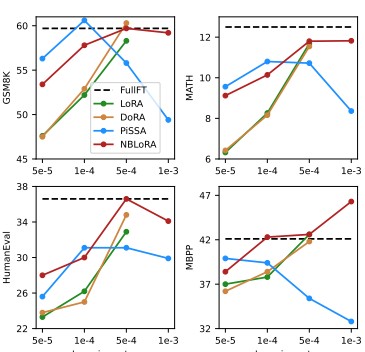

**Figure 3:** Comparison of LoRA, DoRA, PiSSA and NB-LoRA on LLaMA-2-7B with different learning rates.

The training dynamics shown in Figure 4 match the analysis in Section 4.3. LoRA exhibits very small updates $\Delta W$ for an extended period when the learning rate is small. larger learning rate alleviates this issue during the early phase, but may cause training instability. As shown in Section 4.3 (col 4), LoRA shows a relatively small nuclear norm but a much larger spectral norm, indicating that the updates tend to concentrate on a very low-rank subspace, which might be the cause of training instability.

Different from LoRA, PiSSA initializes $\hat{A}$ and $\hat{B}$ based on dominant singular components of the pre-trained weights, leading to significantly larger updates even when the learning rate is small. However, without explicit control, its norm increases substantially for large learning rates, sometimes overwriting useful pretrained structure. This explains its sensitivity to the learning rate observed in Figure 3 and Table 2.

NB-LoRA ensures that $\hat{A}$ and $\hat{B}$ lies on a compact manifold (10), e.g., $\|\hat{A}\|_F^2 + \|\hat{B}\|_F^2 \approx 256 = 2r$ from Figure 4 (Col 3). Hence, NBLoRA can exhibit larger updates $\|\Delta\hat{A}\|_F$ and $\|\Delta\hat{B}\|_F$ than LoRA. On the other hand, because increasing $\|\hat{B}\|_F$ will also deceases $\|\hat{A}\|_F$ simultaneously, the norm of

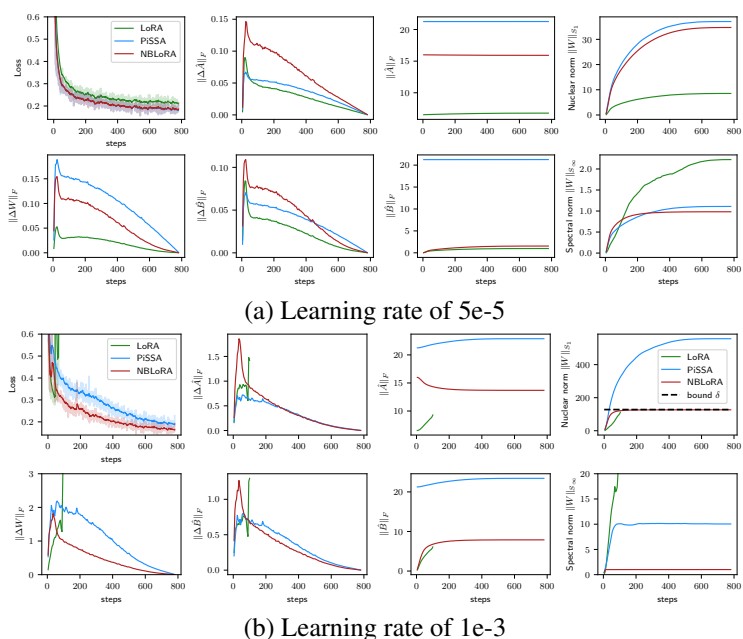

(a) Learning rate of 5e-5

(b) Learning rate of 1e-3

Figure 4: Training dynamics for LoRA, PiSSA and NBLoRA two learning rates.

$W$ remains tightly controlled. With a larger learning rate, NB-LoRA attains the active bound while maintaining stability due to its more uniform singular-value distribution.

**Scalability to Larger Models.** We trained NB-LoRA to LoRA and PiSSA on the LLaMA-3-70B model for GSM8K and compared them in terms of computational resources, accuracy, and learning-rate robustness. In Table 1 it can be seen that NB-LoRA achieved the highest accuracy overall. It uniformly outperformed PiSSA, while standard LoRA achieved good performance for low learning rates but was unstable for larger learning rates. NB-LoRA required slightly more computational resources than LoRA and PiSSA: ∼6% more memory and ∼9% longer training time. More comparison on computation cost can be found in Section E.

| Method | Learning Rate | | | | Computation | |
| --- | --- | --- | --- | --- | --- | --- |
| | 2e-5 | 5e-5 | 1e-4 | 5e-4 | GPU Mem. | Train Time |
| LoRA | 86.0 | 86.2 | 86.2 | failed | 65.57GB | 169m |
| PiSSA | 85.7 | 83.6 | 79.0 | 41.8 | 65.57GB | 170m |
| NB-LoRA | 85.5 | **87.1** | 85.4 | 83.3 | 69.15GB | 185m |

Table 1: GSM8K accuracy of LoRA, PiSSA and NB-LoRA on LLaMA-3-70B with different learning rates.

**Hyperparameter Robustness.** Table 2 compiles the results of a comprehensive sweep across tasks, base models and learning rates, comparing NB-LoRA to LoRA, DoRA, and PiSSA in terms of their robustness to these variations (see table caption for details). While different methods were competitive for different particular scenarios, when averaging across models and tasks NB-LoRA is clearly superior.

**Comparison with DeLoRA.** Figure 5 compares NB-LoRA with DeLoRA (Bini et al., 2025) with $\delta$ set to 10, 20, and free (see Sections 4 and D for discussion). As shown in Figure 5, NB-LoRA achieves substantially larger parameter updates ($\Delta\hat{A}, \Delta\hat{B}, \Delta W$) than DeLoRA—even though NB-LoRA uses a smaller learning rate (1e-3 vs. 5e-3). Making $\delta$ learnable alleviates this limitation to some extent, but it eliminates the norm-bound guarantee, and the GSM8K accuracy of DeLoRA is still lower than NB-LoRA.

### 5.2 VIT FINE-TUNING

The main goal is to explore the utility of norm bounds in preventing catastrophic model forgetting (McCloskey & Cohen, 1989; French, 1999; Wang et al., 2024). Our hypothesis is that tight control

| Base Model | | Mistral-7B-v0.1 | | | | LLaMA-3-8B | | | | LLaMA-2-13B | | | | Model Avg. | | | |
|---|---|---|---|---|---|---|---|---|---|---|---|---|---|---|---|---|---|---|
| Method | | Lo | Do | Pi | NB | Lo | Do | Pi | NB | Lo | Do | Pi | NB | Lo | Do | Pi | NB |
| Math | min | 43.9 | 44.2 | 42.2 | 42.0 | 49.5 | 49.6 | 37.9 | 47.8 | 35.1 | 35.1 | 36.8 | 38.1 | 42.9 | **43.0** | 38.9 | 42.6 |
| | max | 49.1 | 48.1 | 47.0 | 47.9 | 51.5 | 51.8 | 52.0 | 52.9 | 41.8 | 41.0 | 40.4 | 42.2 | 47.4 | 47.0 | 46.5 | **47.7** |
| | avg | 47.2 | 46.8 | 45.4 | 46.0 | 50.5 | 50.9 | 45.6 | 50.3 | 38.9 | 38.8 | 38.7 | 40.4 | 45.5 | 45.5 | 43.2 | **45.6** |
| Code | min | 52.4 | 53.7 | 52.9 | 53.6 | 56.3 | 56.5 | 44.4 | 57.4 | 42.5 | 42.5 | 40.1 | 44.0 | 50.4 | 50.9 | 45.8 | **51.7** |
| | max | 57.8 | 59.2 | 59.0 | 59.7 | 63.2 | 62.6 | 63.0 | 68.1 | 46.6 | 47.2 | 45.6 | 49.4 | 55.9 | 56.4 | 55.9 | **59.1** |
| | avg | 56.1 | 57.0 | 56.0 | 57.5 | 60.3 | 60.5 | 52.6 | 62.0 | 44.7 | 45.0 | 43.9 | 47.6 | 53.7 | 54.2 | 50.8 | **55.7** |
| Task Avg. | min | 48.1 | **48.9** | 47.5 | 47.8 | **52.9** | 53.1 | 41.2 | 52.6 | 38.8 | 38.8 | 38.5 | **41.0** | 46.6 | 46.9 | 42.4 | **47.2** |
| | max | 53.5 | 53.7 | 53.0 | **53.8** | 57.4 | 57.2 | 57.5 | **60.5** | 44.2 | 44.1 | 43.0 | **45.8** | 51.7 | 51.7 | 51.2 | **53.4** |
| | avg | 51.7 | **51.9** | 50.7 | 51.8 | 55.4 | 55.7 | 49.1 | **56.2** | 41.8 | 41.9 | 41.3 | **44.0** | 49.6 | 49.8 | 47.0 | **50.6** |

Table 2: Fine-tuning three base models based on LoRA (Lo), DoRA (Do), PiSSA (Pi) and NB-LoRA (NB) over different learning rates ($\{1e\text{-}5, 5e\text{-}5, 1e\text{-}4, 2e\text{-}4\}$ for Mistral and $\{5e\text{-}5, 1e\text{-}4, 5e\text{-}4, 7e\text{-}4\}$ for LLaMA). We report the minimum, maximum and averaged test results, where the metrics for math and coding are $\frac{1}{2}(\text{GSM8K} + \text{MATH})$ and $\frac{1}{2}(\text{HumanEval} + \text{MBPP})$, respectively.

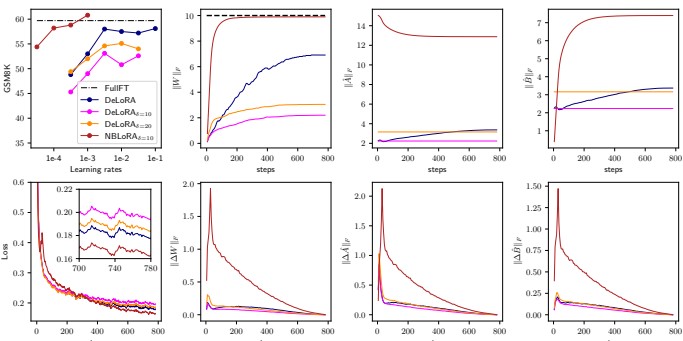

Figure 5: Training dynamics comparison Between NB-LoRA and DeLoRA.

of the adaption norm will prevent loss of performance on the pre-trained model as per the analysis in Section 3, while still enabling good adaptation performance.

**Adaptation vs Forgetting** We perform experiments (Bafghi et al., 2024) on ViT-B/16 model (Dosovitskiy et al., 2020), which is pre-trained on ImageNet-21k (Deng et al., 2009) and then fine-tuned to ImageNet-1k. For the proposed NB-LoRA, we choose the norm bound as $\delta = \gamma \|W_p\|_{S_p}$, where the ratio $\gamma$ is a hyper-parameter. Similar to the setup in Kopiczko et al. (2024a), we adapt $Q, V$ matrices and learn the classification head for the Street View House Number (SVHN) dataset. Here we report the results for NB-LoRA using nuclear norm with bound ratio of $\gamma$ between 0.1 and 1.6, see Section H for additional results with different setups and datasets including CIFAR-100 (Krizhevsky et al., 2009) and Food-101 (Bossard et al., 2014).

The **metric for model forgetting** is the test accuracy of the fine-tuned model on the source dataset: ImageNet-1k, which can be compared against performance on the target dataset. As shown in Figure 6(Left), the linear adapter (i.e. just learning the classification head) avoids forgetting of the source but has poor performance on the target set. In contrast, LoRA, DoRA and PiSSA achieve high adaptation performance to the target data set, but with a dramatic loss of performance on the source data set (from around 80% to less than 10%). VeRA and NB-LoRA can both achieve a good balance of both, but NB-LoRA outperforms in terms of both source and target performance. It can also be seen that tuning of $\gamma$ allows a trade-off between source and target performance.

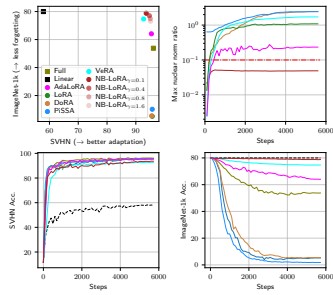

Figure 6: Adaptation to a target dataset vs forgetting of a source dataset.

Figure 6 shows the evolution of source and target accuracy vs training steps. All models (except linear) perform quite similarly in terms of adaptation to the target, whereas on the source dataset NB-LoRA (shown with $\gamma = 0.1$) maintains high accuracy throughout training, while most other methods

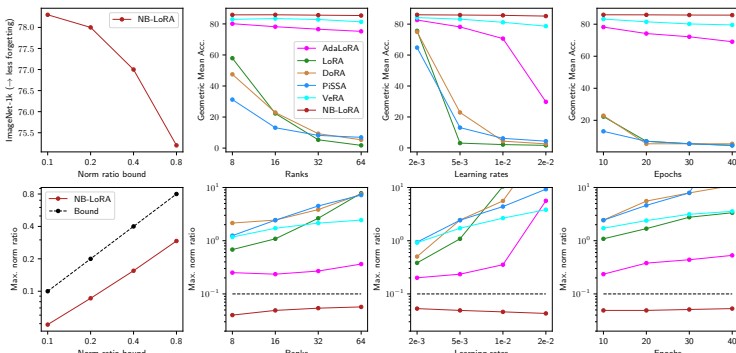

Figure 7: Analysis of hyperparameter robustness of different methods in terms of geometric mean of source (ImageNet-1k) and target (SVHN) dataset accuracies.

quickly forget source performance. For PiSSA, DoRA, and LoRA the source performance drops significantly before target accuracy has converged, so early stopping can not solve the problem. Figure 6 (top right) plots the maximum nuclear norm ratio of the models. NB-LoRA remains below the bound, while several others are more than an order of magnitude larger.

As seen in the top left of Figure 7, when the norm budget $\delta$ of NB-LoRA increases, the corresponding adapter norm also increases and the model forgets more on the source task (ImageNet-1k). Similar trends are observed in other adapters. However, across methods, this relationship does not necessarily hold—for example, VeRA may have larger norms but less forgetting than AdaLoRA. Due to its tight norm control, NB-LoRA consistently exhibits substantially less forgetting than other methods while maintaining good adaptation performance.

## 5.3 SUBJECT-DRIVEN IMAGE GENERATION

Our goal of this experiment is to demonstrate that with tight norm control, NB-LoRA can prevent overfitting for downstream tasks in the low-data regime.

**Task description.** Following Qiu et al. (2023), we evaluate the proposed method in the Deam-Booth setting (Ruiz et al., 2023). We fine-tune Stable Diffusion (Rombach et al., 2022) to contextualize a subject shown in a small set of images together with a given prompt containing a unique token. Following the DreamBooth (Ruiz et al., 2023), we train and evaluate on generating 25 subjects, each of which corresponds to 30 prompts.

**Comparison study.** We conducted comparison experiments of LoRA, DoRA, and NBLoRA with the same learning rate (2e-6) and training horizon (1600 steps). The generated images are evaluated via three crucial aspects: subject fidelity (DINO (Caron et al., 2021), CLIP-I (Radford et al., 2021)), textual prompt fidelity (CLIP-T (Radford et al., 2021)) and sample diversity (LPIPS Zhang et al. (2018)). Table 3 reports the results at the training step where each method achieves its highest DINO score. LoRA and its variants yield high fidelity scores but low prompt fidelity and sample diversity, see the qualitative comparison in Figure 8. Interestingly, the norm bound behaves like a regularization factor. That is, as the bound decreases, the fidelity metrics decrease. Meanwhile, the prompt fidelity and diversity metric increases, indicating the less forgetting for the pretrained model.

| Method | DINO↑ | CLIP-I↑ | CLIP-T↑ | LPIPS↑ |
|---|---|---|---|---|
| Real Images | 0.764 | 0.890 | - | 0.562 |
| LoRA$_{r=16}$ | **0.723** | **0.836** | 0.218 | 0.703 |
| DoRA$_{r=16}$ | 0.718 | 0.834 | 0.217 | 0.704 |
| PiSSA$_{r=16}$ | 0.720 | 0.835 | 0.218 | 0.704 |
| NB-LoRA$_{p=1,\delta=12}$ | 0.702 | 0.816 | 0.238 | 0.717 |
| NB-LoRA$_{p=1,\delta=6}$ | 0.648 | 0.781 | 0.261 | 0.743 |
| NB-LoRA$_{p=1,\delta=4}$ | 0.593 | 0.750 | 0.274 | **0.761** |
| NB-LoRA$_{p=2,\delta=1.8}$ | 0.657 | 0.790 | 0.258 | 0.740 |
| NB-LoRA$_{p=2,\delta=1.5}$ | 0.647 | 0.779 | 0.261 | 0.743 |
| NB-LoRA$_{p=2,\delta=1.0}$ | 0.592 | 0.748 | **0.275** | **0.761** |
| NB-LoRA$_{p=\infty,\delta=0.9}$ | 0.709 | 0.823 | 0.235 | 0.717 |
| NB-LoRA$_{p=\infty,\delta=0.5}$ | 0.670 | 0.795 | 0.253 | 0.734 |
| NB-LoRA$_{p=\infty,\delta=0.25}$ | 0.594 | 0.750 | **0.275** | **0.761** |

Table 3: Quantitative comparison of subject fidelity (DINO, CLIP-I), prompt fidelity (CLIP-T) and diversity metric (LPIPS).

**Prolonged training.** We further investigate the behavior of different methods via the weight norm changes during fine-tuning. Figure 9 shows that LoRA and its variants (DoRA, PiSSA) continuously

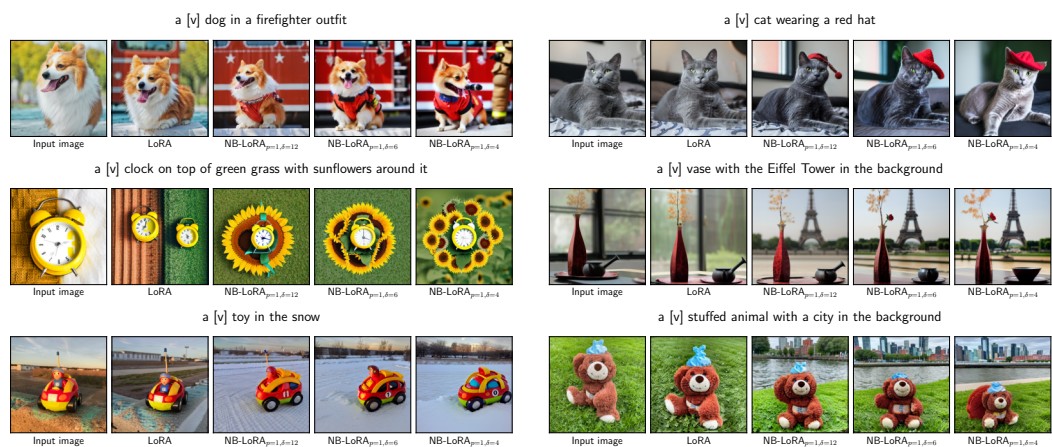

Figure 8: Qualitative comparison of subject-driven generation among LoRA and NB-LoRA with different nuclear norm bounds.

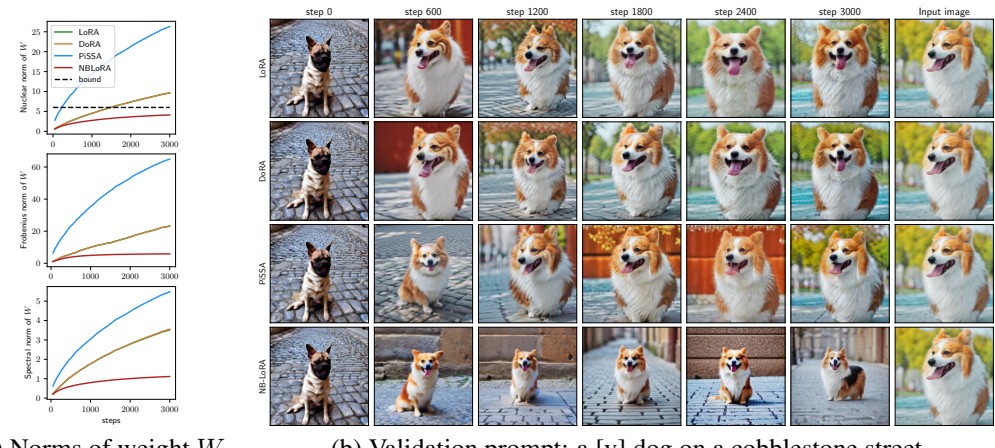

(a) Norms of weight $W$      (b) Validation prompt: a [v] dog on a cobblestone street

Figure 9: (Left) Norms of fine-tuned weights as a function of training steps. (Right) Qualitative examples show that LoRA, DoRA and PiSSA exhibits significant overfitting compared with NB-LoRA, which maintains better prompt fidelity and diversity.

depart from the pretrained weights as the norm increase substantially, which leads to overfitting issues due to tiny dataset (5 6 images). Due to the tight bound control, NB-LoRA exhibits prolonged training robustness and effectively avoid model overfitting.

# 6 LIMITATIONS

Although we show norm-controlled low-rank adaption is useful, there are fine-tuning tasks that *do* require high–spectral-norm update in order to encode the new knowledge from downstream dataset. For such tasks, vanilla LoRA may indeed be more suitable than NB-LoRA.

# 7 CONCLUSION

In this paper we propose a norm-bounded low-rank adaptation (NB-LoRA) for model fine tuning. In particular, we introduce a new parameterization which is smooth and complete, i.e. it covers all matrices of a specified rank and singular value bounds. The proposed parameterization address some issues related to the initialization of LoRA and its impact on learning rate, and can also mitigate the tendency of LoRA to forget source model performance and overfit small target dataset.

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

## A    KEY TECHNICAL LEMMAS

Here we present some key lemmas which are used in our proofs later.

**Lemma A.1.** *For any $Q \in \mathbb{R}^{n \times n}$, there exists a diagonal matrix $P$ with $P_{jj} \in \{-1, 1\}$ such that $I + PQ^\top$ is invertible.*

*Proof.* Let $e_k, q_k$ be the $k$th column of $I$ and $Q$, respectively. We construct $A_k$ via

$$A_k^{-1} = A_{k-1}^{-1} - \frac{s_k A_{k-1}^{-1} e_k q_k^\top A_{k-1}^{-1}}{1 + s_k q_k^\top A_{k-1}^{-1} e_k}, \tag{14}$$

where $A_0 = I$ and $s_k = \text{sign}\big(v_k^\top A_{k-1}^{-1} e_k\big)$ with $\text{sign}(0) = 1$. From Sherman-Morrison formula, $A_k$ is well-defined (i.e., invertible) and satisfies $A_k = A_{k-1} + s_k e_k q_k^\top$. By taking $P = \text{diag}(s_1, \ldots, s_n)$, we have $A_n = I + \sum_{k=1}^n s_k e_k q_k^\top = I + PQ^\top$ is also invertible. $\square$

**Lemma A.2.** *Let $G \in \mathbb{R}^{r \times r}$ and $H \in \mathbb{R}^{s \times r}$ such that $G^\top G + H^\top H = I$. Then,*

$$\begin{bmatrix} G \\ H \end{bmatrix} = \text{Cayley}\left(\begin{bmatrix} X \\ Y \end{bmatrix}\right) = \begin{bmatrix} (I - Z)(I + Z)^{-1} \\ -2Y(I + Z)^{-1} \end{bmatrix} \tag{15}$$

*for some $X \in \mathbb{R}^{r \times r}$ and $Y \in \mathbb{R}^{s \times r}$ if and only if $I + G$ is invertible.*

*Proof.* From the Cayley transformation (7) we have the following relationships:

$$G = (I - Z)(I + Z)^{-1}, \quad H = -2Y(I + Z)^{-1}, \quad Z = X - X^\top + Y^\top Y. \tag{16}$$

(**if**). From the above equation we have $I + G = (I + Z)^{-1}$ invertible.

(**only if**). The proof is constructive, i.e., finding $X, Z \in \mathbb{R}^{r \times r}$ and $Y \in \mathbb{R}^{s \times r}$ satisfying (16). We consider a candidate solution as follows:

$$Z = (I + G)^{-1}(I - G), \quad Y = -\frac{1}{2}H(I + Z), \quad X = \frac{1}{2}Z. \tag{17}$$

It is easy to check that the above solution satisfies the first two equations in (16). We now verify the last equation as follows:

$$Z + X^\top - X - Y^\top Y = \frac{1}{2}(Z + Z^\top) - Y^\top Y$$

$$= \frac{1}{2}[(I + G)^{-1}(I - G) + (I - G^\top)(I + G^\top)^{-1}] - (I + G^\top)^{-1}H^\top H(I + G)^{-1}$$

$$= \frac{1}{2}[(I - G)(I + G)^{-1} + (I + G^\top)^{-1}(I - G^\top)] - (I + G^\top)^{-1}H^\top H(I + G)^{-1}$$

$$= \frac{1}{2}(I + G^\top)^{-1}[(I + G^\top)(I - G) + (I - G^\top)(I + G) - 2H^\top H](I + G)^{-1}$$

$$= (I + G^\top)^{-1}[I - G^\top G - H^\top H](I + G)^{-1} = 0,$$

where the second line is due to that $(I + G)^{-1}$ and $(I - G)$ are commutative. $\square$

**Lemma A.3.** *Let $A \in \mathbb{R}^{r \times m}$ and $B \in \mathbb{R}^{r \times n}$ with $AA^\top + BB^\top = I$. Then, there exist a diagonal matrix $P \in \mathbb{R}^{r \times r}$ with $P_{jj} \in \{-1, 1\}$ and $\tilde{A} \in \mathbb{R}^{r \times m}, \tilde{B} \in \mathbb{R}^{r \times n}$ satisfying*

$$[PA \quad PB]^\top = \text{Cayley}\left([\tilde{A} \quad \tilde{B}]^\top\right). \tag{18}$$

*Proof.* From the assumption we have that $\begin{bmatrix} A^\top \\ B^\top \end{bmatrix}$ is a tall matrix, i.e., $r \leqslant m + n$. We then take the partition $\begin{bmatrix} A^\top \\ B^\top \end{bmatrix} = \begin{bmatrix} \bar{G} \\ \bar{H} \end{bmatrix}$ with $\bar{G} \in \mathbb{R}^{r \times r}$ and $\bar{H} \in \mathbb{R}^{(m+n-r) \times r}$. We introduce $G = \bar{G}P$ and $H = \bar{H}P$, where $P$ is a diagonal matrix with $P_{jj} \in \{-1, 1\}$. Then, we can obtain

$$G^\top G + H^\top H = P(\bar{G}^\top \bar{G} + \bar{H}^\top \bar{H})P = P(AA^\top + BB^\top)P = P^2 = I$$

for all diagonal such $P$. From Theorem A.1, we can pick a particular $P$ such that $I + G = I + \bar{G}P$ is invertible. We then follow Theorem A.2 to compute $X \in \mathbb{R}^{r \times r}$ and $Y \in \mathbb{R}(m + n - r) \times r$ satisfying (15). Finally, we take the partition $[X^\top \quad Y^\top] = [\tilde{A} \quad \tilde{B}]$. $\square$

# B PROOF OF THEOREM 4.2

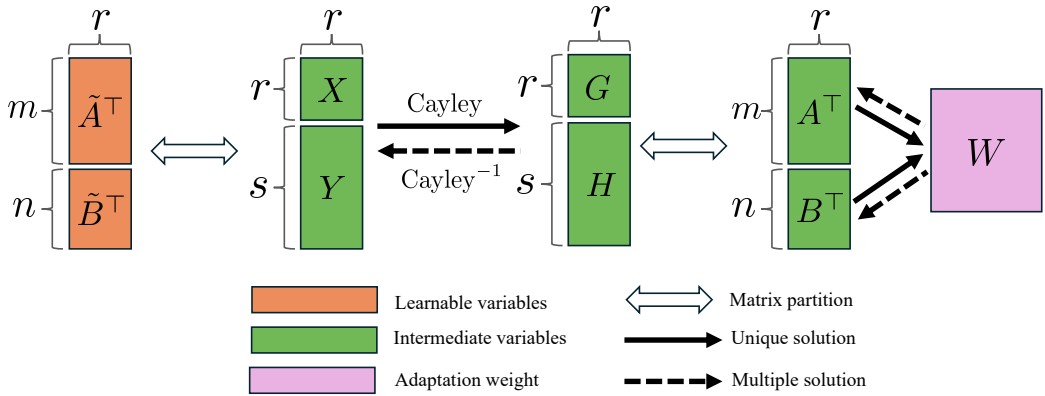

Figure 10: Diagram of NB-LoRA parameterization.

The proof includes two parts: I) $W = \mathcal{W}(\tilde{A}, \tilde{B}) \in \mathbb{W}_S$ for any $\tilde{A} \in \mathbb{R}^{r \times m}$ and $\tilde{B} \in \mathbb{R}^{r \times n}$; II) for any $W \in \mathbb{W}_S$, there exists a pair of $\tilde{A} \in \mathbb{R}^{r \times m}$ and $\tilde{B} \in \mathbb{R}^{r \times n}$ such that $W = \mathcal{W}(\tilde{A}, \tilde{B})$.

**Part I** It is obvious that $\mathrm{rank}(W) \leqslant r$. The $j$th singular value of $W$ satisfies

$$\sigma_j(W) = 2\sigma_j(\underbrace{B^\top S^{\frac{1}{2}}}_{Q^\top} \underbrace{S^{\frac{1}{2}} A}_{K}) \leqslant \sigma_j\left(QQ^\top + KK^\top\right) = \sigma_j(S^{\frac{1}{2}}(\underbrace{AA^\top + BB^\top}_{I})S^{\frac{1}{2}}) = \sigma_j(S) \quad (19)$$

where the inequality is the matrix arithmetic-geometric mean inequality (Bhatia & Kittaneh, 1990; Bhatia, 2013), and the last equality follows by the Cayley transformation.

**Part II** Without loss of generality, we assume that the diagonal elements of $S$ is in descending order, i.e., $\sigma_j(S) = S_{jj}$ for $j = 1, \ldots, r$. Since $W$ has maximally $r$ non-zero singular values, we can take the reduced SVD decomposition $W = U_w \Sigma_w V_w^\top$ where $U_w \in \mathbb{R}^{m \times r}, V_w \in \mathbb{R}^{n \times r}$ are semi-orthogonal, and the positive diagonal matrix $S_w \in \mathbb{R}^{r \times r}$. We now consider the following candidates for $A, B$:

$$A = P\Sigma_a V_w^\top, \quad B = P\Sigma_b U_w^\top, \quad (20)$$

where $P \in \mathbb{R}^{r \times r}$ is a diagonal matrix with $P_{jj} \in \{-1, 1\}$, and $\Sigma_a, \Sigma_b \in \mathbb{R}^{r \times r}$ are positive diagonal matrices. The first constraint for $A$ and $B$ is that $\begin{bmatrix} A & B \end{bmatrix}^\top$ is semi-orthogonal since it is an output of the Cayley transformation. Thus, we have

$$I = AA^\top + BB^\top = P(\Sigma_a^2 + \Sigma_b^2)P^\top \implies \Sigma_a^2 + \Sigma_b^2 = I. \quad (21)$$

The second constraint for $A, B$ is $W = 2B^\top SA$, which implies

$$U_w \Sigma_w V_w^\top = 2U_w \Sigma_a P^\top SP \Sigma_b V_w^\top = U_w(2\Sigma_a \Sigma_b S)V_w^\top \implies 2\Sigma_a \Sigma_b = \Sigma_w S^{-1} \quad (22)$$

Eq. (21) and (22) yield a solution of

$$\Sigma_a = \frac{\sqrt{I+J} + \sqrt{I-J}}{2}, \ \Sigma_b = \frac{\sqrt{I+J} - \sqrt{I-J}}{2}. \quad (23)$$

where $J = \Sigma_w S^{-1}$ satisfies $0 \leq J \leq I$ since $S_w \leq S$ for $W \in \mathbb{W}_S$. Note that we need to deal with the case where $S$ is not full rank, i.e., there exists an $k < r$ such that $S_{kk} = 0$ and $S_{k-1,k-1} > 0$. Since $0 \leq \Sigma_w \leq S$, we have $\Sigma_{ii} = 0$ for all $i \geqslant k$ and simply take $J_{ii} = 1$. It is easy to verify that Equations (21) - (23) still hold. Finally, Theorem A.3 shows that we can recover $\tilde{A}, \tilde{B}$ from $A, B$ by picking a proper $P$ in (20) based on Theorem A.1.

| Method | Peak Mem. | Train Time | GSM8K Acc. |
|---|---|---|---|
| AutoDiff | 69.52GB | 23m51s | 58.0 |
| Custom | 67.19GB | 22m40s | 57.8 |

Table 4: Computation comparison for training NB-LoRA with AutoDiff and custom backward step.

## C  CUSTOM BACKWARD FOR CAYLEY TRANSFORMATION

We first rewrite the forward computation of Cayley transformation $(X, Y) \to (G, H)$ as follows:

$$Z = X - X^\top + Y^\top Y, \quad M = I + Z, \quad W = M^{-1}, \quad G = (I - Z)W, \quad H = -2YW \quad (24)$$

where $G, X, Z, M, W \in \mathbb{R}^{r \times r}$ and $H, Y \in \mathbb{R}^{s \times r}$. We provide a custom backward $(\nabla_G, \nabla_H) \to (\nabla_X, \nabla_Y)$ with $\nabla_A = (\partial \ell / \partial A)^\top$ for (24) as follows:

$$\begin{bmatrix} \tilde{\nabla}_G \\ \tilde{\nabla}_H \end{bmatrix} = \begin{bmatrix} \nabla_G \\ \nabla_H \end{bmatrix} W^\top, \quad S_Z = \begin{bmatrix} I + G \\ H \end{bmatrix}^\top \begin{bmatrix} \tilde{\nabla}_G \\ \tilde{\nabla}_H \end{bmatrix},$$

$$\nabla_X = S_Z^\top - S_Z, \quad \nabla_Y = -\frac{1}{2} HM(S_Z^\top + S_Z) - 2\tilde{\nabla}_H, \quad (25)$$

where $W, M, G, H$ can be reused from the Cayley forward step. Note that when applying AutoDiff to (24), it is necessary to store the input $X, Y$, output $G, H$ as well as some intermediate steps, which requires more memory than our custom backward step (25) since $Y \in \mathbb{R}^{s \times r}$ is much larger than $W, M \in \mathbb{R}^{r \times r}$. In our approach, we can recover $Y$ from other stored variables, i.e., $Y = -\frac{1}{2} HM$. To give detail derivation for (25), we first differentiate the forward step (24):

$$\mathrm{d}Z = \mathrm{d}X - \mathrm{d}X^\top + Y^\top \mathrm{d}Y + \mathrm{d}Y^\top Y, \quad \mathrm{d}W = -W\mathrm{d}ZW,$$

$$\mathrm{d}G = -\mathrm{d}ZW + (I - Z)\mathrm{d}W, \quad \mathrm{d}H = -2\mathrm{d}YW - 2Y\mathrm{d}W. \quad (26)$$

The differential of loss function $\mathrm{d}\ell$ satisfies

$$\mathrm{d}\ell = \mathrm{Tr}\left(\nabla_X^\top \mathrm{d}X\right) + \mathrm{Tr}\left(\nabla_Y^\top \mathrm{d}Y\right) = \mathrm{Tr}\left(\nabla_G^\top \mathrm{d}G\right) + \mathrm{Tr}\left(\nabla_H^\top \mathrm{d}H\right). \quad (27)$$

By further substituting (26) into (27), we have

$$\mathrm{Tr}\left(\nabla_G^\top \mathrm{d}G\right) + \mathrm{Tr}\left(\nabla_H^\top \mathrm{d}H\right)$$

$$= -\mathrm{Tr}\left(\nabla_G^\top \mathrm{d}ZW + \nabla_G^\top (I - Z)W\mathrm{d}ZW\right) - 2\mathrm{Tr}\left(\nabla_H^\top \mathrm{d}YW - \nabla_H^\top YW\mathrm{d}ZW\right)$$

$$= -\mathrm{Tr}\left(W(\nabla_G^\top + \nabla_G^\top(I - Z)W - 2\nabla_H^\top YW)\mathrm{d}Z\right) - \mathrm{Tr}\left(2W\nabla_H^\top \mathrm{d}Y\right)$$

$$= -\mathrm{Tr}\left(W(\nabla_G^\top + \nabla_G^\top G + \nabla_H^\top H)\mathrm{d}Z\right) - \mathrm{Tr}\left(2W\nabla_H^\top \mathrm{d}Y\right)$$

$$= -\mathrm{Tr}\left(S_Z^\top \mathrm{d}Z\right) - \mathrm{Tr}\left(2W\nabla_H^\top \mathrm{d}Y\right)$$

$$= -\mathrm{Tr}\left((S_Z^\top - S_Z)\mathrm{d}X\right) - \mathrm{Tr}\left(((S_Z + S_Z^\top)Y^\top + 2W\nabla_H^\top)\mathrm{d}Y\right) = \mathrm{Tr}\left(\nabla_X^\top \mathrm{d}X\right) + \mathrm{Tr}\left(\nabla_Y^\top \mathrm{d}Y\right)$$

which yields the custom backward step (25) by substituting $Y = -\frac{1}{2}HM$. As shown in Table 4, the custom backward pass can save both GPU memory and training time.

## D  CONNECTIONS BETWEEN DELORA AND NB-LORA

DeLoRA (Bini et al., 2025) is a fine-tuning method which can control both rank and Frobenius norm bound of weight adaptation $W$. Specifically, DeLoRA takes the form of

$$W = \frac{\delta}{r} B^\top \Xi A := \frac{\delta}{r} B^\top \mathrm{diag}(|b_i|_2 \cdot |a_i|_2) A \quad (28)$$

where $a_i, b_i$ are the $i$th row of $A \in \mathbb{R}^{r \times n}$ and $B \in \mathbb{R}^{r \times m}$, respectively. The above parameterization can be rewritten as sum of NB-LoRA matrices with both rank and norm bound of 1:

$$W = \frac{\delta}{r} \sum_{i=1}^{r} 2 \left(\frac{b_i}{\sqrt{2}|b_i|_2}\right)^\top \left(\frac{a_i}{\sqrt{2}|a_i|_2}\right) = \frac{\delta}{r} \sum_{i=1}^{r} 2\bar{b}_i^\top \bar{a}_i = \frac{\delta}{r} \sum_{i=1}^{r} \bar{W}_i,$$

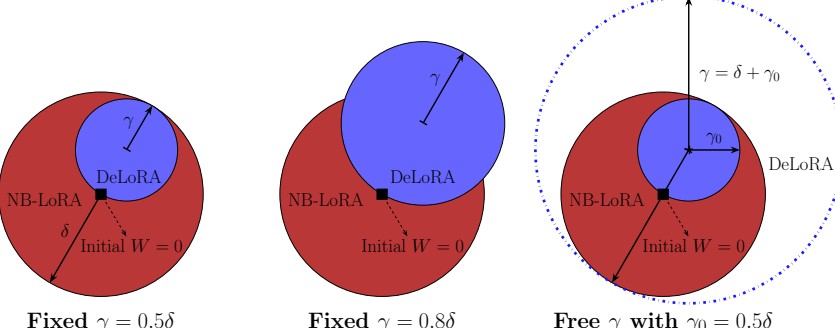

**Fixed $\gamma = 0.5\delta$**    **Fixed $\gamma = 0.8\delta$**    **Free $\gamma$ with $\gamma_0 = 0.5\delta$**

Figure 11: Visualization of the reachable sets $\mathbb{W}_{\text{NBLoRA}}$ (red) and $\mathbb{W}_{\text{DeLoRA}}$ (blue). (Left) With frozen scaling factor $\gamma = 0.5\delta$, DeLoRA provides the same certified norm bound as NB-LoRA while $\mathbb{W}_{\text{DeLoRA}}$ is much smaller than $\mathbb{W}_{\text{NBLoRA}}$. (Middle) Further increasing the fixed $\gamma$ can enlarge $\mathbb{W}_{\text{DeLoRA}}$ but it does not cover $\mathbb{W}_{\text{NBLoRA}}$. (Right) DeLoRA can cover $\mathbb{W}_{\text{NBLoRA}}$ if $\gamma$ is free and sufficiently large, i.e., $\gamma \geqslant \delta + \gamma_0$. However, its norm bound $\delta + 2\gamma_0$ is much larger than NB-LoRA.

where $\begin{bmatrix} \bar{a}_i & \bar{b}_i \end{bmatrix}$ is a set of of decoupled unit vectors. By Theorem 4.2 we have that $\|\bar{W}_i\|_F \leqslant 1$ and $\|W\|_F \leqslant \delta/r \sum_{i=1}^{r} \|\bar{W}_i\|_F \leqslant \delta$. NB-LoRA in (6) also has a similar representation:

$$W = 2B^\top S A = \sum_{i=1}^{r} s_i (2\hat{b}_i^\top \hat{a}_i) = \sum_{i=1}^{r} s_i \hat{W}_i.$$

Different from DeLoRA, $\begin{bmatrix} \hat{a}_i & \hat{b}_i \end{bmatrix}$ is a set of coupled unit vectors as they are orthogonal to each other. This coupling behavior allows us to specify the bound for each singular value of $W$, providing tight control of a wide family of matrix norms.

Another main difference is model initialization. Since it is not straightforward to initialize $A, B$ satisfying $W = 0$ for (28), the residual-type initialization (Meng et al., 2024) is adopted, leading to different reachable sets $\mathbb{W}_{\text{NBLoRA}}$ and $\mathbb{W}_{\text{DeLoRA}}$ when an explicit norm bound is specified. From Theorem 4.2 we have that $\mathbb{W}_{\text{NBLoRA}}$ covers the feasible region of $W$ with norm bound of $\delta$. Moreover, the initial point $W = 0$ of NB-LoRA lies at the center of the feasible region, allowing searching for all directions, see Figure 11. Due to the residual type initialization, DeLoRA requires a fixed $\gamma = \delta/2$ to ensure the same norm bound guarantee, see the left of Figure 11. Since its initial $W$ lies at the boundary of $\mathbb{W}_{\text{DeLoRA}}$, the searching directions of DeLoRA are constrained in certain ranges that depend on the random initial guess. Although these issues can be resolved by making $\gamma$ learnable, DeLoRA allows an unbounded Frobenius norm and needs a larger bound to cover the range of NB-LoRA, see the right of Figure 11.

# E    COMPOTATION COST COMPARISON

Table 5 compares computational costs against two methods, PiSSA and DoRA, across different ranks on LLaMA 2-7B. Due to the extra reparameterization layer, NB-LoRA takes slightly more GPU memory and training time. NB-LoRA with the largest rank $r = 256$ still takes less computational resources than DoRA with the smallest rank $r = 2$. The main reason is that DoRA requires explicit calculation of the full adaptation matrix, which can be avoided with LoRA and NB-LoRA. Specifically, DoRA (Liu et al., 2024) decouples angular and magnitude components of weight adaptation via

$$W = \underline{m} \frac{(W_p + B^\top A)}{\|W_p + B^\top A\|_c}$$

with $\|\cdot\|_C$ as the column-wise vector norm. Note that the normalization vector $\|W_p + B^\top A\|_c$ requires computing $B^\top A \in \mathbb{R}^{m \times n}$, whose forward computation time could be much larger than $r \times r$-matrix inverse, see Table 6.

| Rank | | 2 | 4 | 8 | 16 | 32 | 64 | 128 | 256 |
|---|---|---|---|---|---|---|---|---|---|
| Training Time | DoRA | 24m46s | 24m33s | 24m03s | 24m04s | 24m05s | 24m02s | 24m18s | 24m53s |
| | PiSSA | 17m44s | 17m35s | 17m09s | 17m10s | 17m12s | 17m15s | 17m32s | 18m09s |
| | NB-LoRA | 18m53s | 18m55s | 18m26s | 18m38s | 18m51s | 19m13s | 20m15s | 22m40s |
| Peak GPU Mem. (GB) | DoRA | 102.41 | 102.44 | 102.51 | 102.64 | 102.90 | 103.41 | 104.47 | 106.53 |
| | PiSSA | 60.92 | 60.96 | 61.02 | 61.15 | 61.41 | 61.93 | 62.96 | 65.04 |
| | NB-LoRA | 60.94 | 60.99 | 61.08 | 61.28 | 61.67 | 62.45 | 64.05 | 67.19 |

Table 5: Computation comparison of DoRA, PiSSA and NB-LoRA with rank choice from 2 to 256. Experiments are conducted with 4 H200 GPUs.

| Matrix Operation | 2 | 4 | 8 | 16 | 32 | 64 | 128 | 256 |
|---|---|---|---|---|---|---|---|---|
| $B^\top A \in \mathbb{R}^{m \times n}$ | 314.1±2.5 | 311.9±1.6 | 312.1±2.4 | 310.7±2.0 | 288.3±2.0 | 323.9±1.9 | 421.9±2.0 | 792.6±1.1 |
| $M^{-1} \in \mathbb{R}^{r \times r}$ | 29.2±0.9 | 30.7±0.9 | 35.1±1.4 | 48.3±0.9 | 72.9±1.1 | 97.0±2.0 | 169.7±1.4 | 361.3±1.7 |

Table 6: Computation time ($\mu$s) of the rank-$r$ matrix $B^\top A \in \mathbb{R}^{m \times n}$ in DoRA and $M^{-1} \in \mathbb{R}^{r \times r}$ in NB-LoRA. We use $m = 4096$, $n = 4m$ and rank $r$ from 2 to 256. Computation time is measured based on 500 samples with 500 warm-up steps on RTX4090.

| Design choice | Method | $W$ formulation |
|---|---|---|
| | LoRA | $\frac{\alpha}{r} B^\top A$ |
| +(Cayley transform) | NB-LoRA with $\|W\|_{S_\infty} \leqslant \delta$ | $2\delta B^\top A$ with $(A, B) = \text{Cayley}(\tilde{A}, \tilde{B})$ |
| +(learnable scaling) | NB-LoRA with $\|W\|_{S_p} \leqslant \delta$ | $2\delta B^\top S A$ with $S = \text{diag}(s)$ and $\|s\|_p \leqslant \delta$ |

Table 7: Summary of incremental design choices from LoRA to NB-LoRA.

| Rank | Method | Learning Rate | | |
|---|---|---|---|---|
| | | 1e-4 | 5e-4 | 1e-3 |
| 128 | LoRA | 52.8 | 58.3 | failed |
| | NB-LoRA (spectral) | 57.7 | **60.5** | 60.0 |
| | NB-LoRA (nuclear) | 57.8 | 59.7 | 59.2 |
| 16 | LoRA | 43.2 | 55.8 | 57.5 |
| | NB-LoRA (spectral) | 47.9 | 55.3 | 56.5 |
| | NB-LoRA (nuclear) | 49.4 | **56.8** | 55.6 |

Table 8: We report the GSM8K accuracy for ablation of NB-LoRA on fine-tuning LLaMA-2-7B models with different ranks and learning rates.

# F LLM EXPERIMENTAL DETAILS AND ADDITIONAL RESULTS

**Training Details.** In our LLM experiments, we use the same training setup as Meng et al. (2024); Taori et al. (2023), i.e., AdamW Loshchilov & Hutter (2019) with no weight decay. We use the cosine annealing scheduler with a warm-up ratio of 0.03. The default batch size is 128. We ensure $\alpha = r$ for all adapters, although NB-LoRA does not use this parameter. We choose the norm bound of $\delta = r$ with nuclear norm, which results in the same scaling factor as the other adapters. When Frobenius or spectral norm is used, we set the default bound as $\delta = \sqrt{r}$ and $\delta = 1$, respectively, which also results in the same scaling factor as other adapters. We set `lora_dropout` to 0, and insert the adapters into all linear layers of the base model. We use BFloat16 for both the base model and the adapters.

**Ablation of NB-LoRA Design Choice.** We summarize the incremental design choices that transform LoRA into NB-LoRA in Table 7. We conduct an ablation study on LLaMA-2-7B fine-tuning in Table 8. For a large rank ($r = 128$), NB-LoRA with spectral norm bound achieves slightly better performance, whereas the nuclear norm performs better under a low-rank budget. Both methods yield more robust performance compared to LoRA.

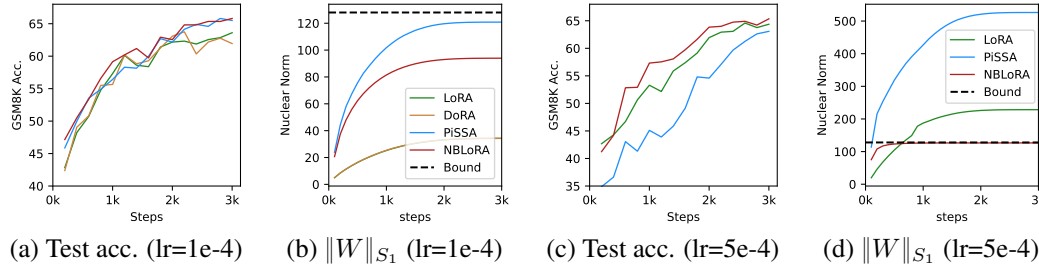

(a) Test acc. (lr=1e-4)  (b) $\|W\|_{S_1}$ (lr=1e-4)  (c) Test acc. (lr=5e-4)  (d) $\|W\|_{S_1}$ (lr=5e-4)

Figure 12: The evaluation accuracy, the nuclear norm bound, loss and grad norm over a full training epoch on MetaMathQA. The norm bound is computed by maximizing over all adaptation modules.

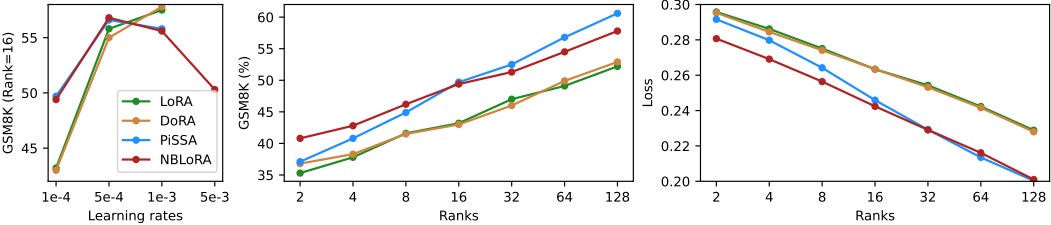

Figure 13: Comparison among LoRA, DoRA, PiSSA and NB-LoRA across ranks from 2 to 128. We also test the learning rate robustness for the case with $r = 16$.

**Robust Performance for Prolong Training.**  We conduct a full epoch training of LLaMA-2-7B on the MetaMathQA dataset. The learning rates are chosen to be 1e-4 and 5e-4, which achieve good performance for different adapters in short horizon training. As shown in Figure 12, we can observe that NB-LoRA consistently outperforms other methods. In particular, NB-LoRA is more stable for the large learning rate, due to the norm saturation on weight adaptation. Meanwhile, DoRA depicts unstable training and PiSSA has poor performance due to excessive increase in weight norm.

**Experiments on Various Ranks.**  Figure 13 explores the impact of rank on LoRA, DoRA, PiSSA and NB-LoRA with learning rate of 1e-4. Under the setup, PiSSA achieves the best GSM8K accuracy. As the rank decrease, the gap between NB-LoRA and PiSSA narrows. And NB-LoRA outperforms PiSSA for low ranks when $r < 16$. NB-LoRA outperforms LoRA and DoRA by approximately 5% across all ranks. We also examine the effect of varying learning rates at rank 16, demonstrating robustness to learning rate choices across different ranks.

**Addition Computation Comparison between DoRA and NB-LoRA.**  We first report the forward computation time of key operations in DoRA and NB-LoRA in Table 6, showing that inverting a small low rank matrix is much computationally cheaper than computing a large low-ran weight matrix.

## G  CHOICE OF NORM BOUND

Here we give a theoretical explanation that for the proposed NB-LoRA approach, the norm bound $\delta$ can be understood as a regularization coefficient. NB-LoRA reparameterizes the following Ivanov regularization problem:

$$\min_W \ \ell(W) \quad \text{s.t.} \quad \|W\| \leqslant \delta.$$

which is closely related to the Tikhonov formulation:

$$\min_W \ \ell(W) + \lambda \|W\|.$$

Under mild assumptions, these two problems are equivalent Oneto et al. (2016). Since NB-LoRA provides a complete parameterization over the feasible set, the constrained problem can be expressed as an unconstrained one without loss of expressivity. Thus, $\delta$ has a similar effect to $\lambda$, which can

| Hyper Param. $\alpha$ | 32 | 64 | 128 | 256 | 512 |
|---|---|---|---|---|---|
| LoRA (lr=5e-4) | 57.9 | 58.3 | 58.3 | **60.3** | 57.8 |
| LoRA (lr=2e-4) | 51.0 | 54.2 | 55.8 | 58.6 | **59.9** |
| LoRA (lr=1e-4) | 48.7 | 52.2 | 54.0 | **57.0** | 56.6 |
| Hyper Param. $\delta$ | 16 | 32 | 64 | 128 | 256 |
| NB-LoRA (lr=5e-4) | 55.8 | 58.2 | **60.6** | 59.7 | 58.8 |
| NB-LoRA (lr=2e-4) | 51.4 | 54.5 | 58.1 | 59.1 | **61.1** |
| NB-LoRA (lr=1e-4) | 51.4 | 54.7 | 57.8 | **60.1** | 59.4 |

Table 9: We report the GSM8K accuracy of fine-tuning LoRA and NB-LoRA with different choices of hyper-parameters.

be verified via the empirical results in Table 9. It is worth wo mention that for Table 2, the default choice of $\alpha$ also shows similar U-shape behavior as the learning rate changes, where doubled or half $\alpha$ can lead to either training instability or substantial performance degradation on average.

## H  ViT Experiments

**Training Details.**  A similar ViT fine-tuning experiments for the model forgetting issue can be found in Bafghi et al. (2024). We take the ViT-B/16 model Dosovitskiy et al. (2020) and insert adaption blocks into the $Q, V$ matrices Kopiczko et al. (2024a). We choose AdamW Loshchilov & Hutter (2019) as the optimizer with default learning rate of 5e-3 and weight decay of 0.01. For the full fine-tuning, we reduce the learning rate to 5e-4. We take one-cycle learning rate scheduler with warm-up ratio of 0.1. We use batch size of 128 for SVHN dataset and 256 for CIFAR-100 and Food-101 dataset.

**Extra results.**  We report the ViT examples with different target datasets: CIFAR-100 and Food-101 in Figure 14. A similar conclusion as the SVHN experiment can be drawn from two datasets.

## I  Subject-Driven Image Generation

In this section we report further details about experiments in Section 5.3. We adopt the code implementation form `examples/boft_dreambooth` in the Hugging Face PEFT library. We tuned the learning rate from 6e-4 (i.e., the default choice in Bini et al. (2025)) to 2e-4, since we observed that LoRA with lr=6e-4 exhibits overfitting at early stage in our experimental setup.

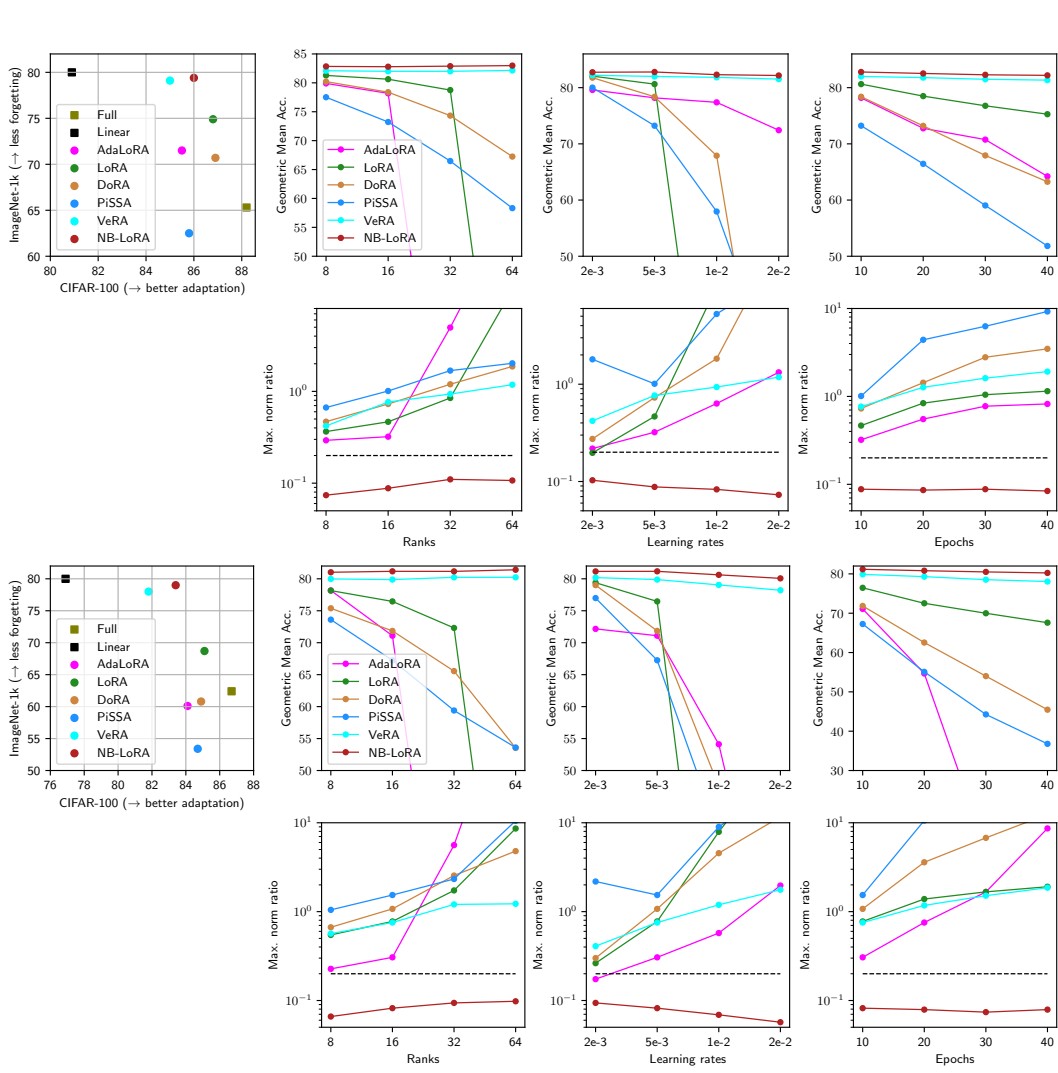

Figure 14: Geometric mean of CIFAR-100 (top) and Food-101 (bottom) with different adapters on various of hyper-parameter setup.

