# OpenReview forum: "Norm-Bounded Low-Rank Adaptation"
_ICLR.cc/2026/Conference — Submitted to ICLR 2026_

### Official Review · Reviewer_smpu · 2025-10-15

**Soundness:** 3
**Presentation:** 3
**Contribution:** 2
**Rating:** 2
**Confidence:** 4

**Summary:**

This paper proposes NB-LoRA (Norm-Bounded Low-Rank Adaptation), a parameterization for parameter-efficient fine-tuning (PEFT) methods that enables explicit and smooth control over both the rank and unitarily invariant matrix norms of low-rank weight adaptations. The authors show that their construction can control the singular value spectrum of the adaptation matrix while maintaining completeness, and provide theoretical insights linking the method to improved initialization, gradient flow, and model forgetting mitigation.

**Strengths:**

1.	The NB-LoRA parameterization allows arbitrarily prescribable bounds on all singular values, enabling control over a variety of unitarily invariant matrix norms (not just Frobenius). Moreover, the authors clearly show the potential benefits of such a method with its insights provided in Section 3.

2.	The methodology is tested across multiple backbone models, including LLaMA-2/3 (up to 70B params), Mistral-7B, and ViT-B/16, and also on diverse datasets (math, code, vision), providing a robust suite of experimental results.

3.	In terms of clarity and presentation, the paper is well-written and easy to follow. The motivation, methodology, and experimental setup are generally explained in a clear and organized manner, making the overall contribution accessible to readers.

**Weaknesses:**

1. Novelty: The core idea of controlling the norm of updated LoRA blocks has been extensively explored in prior works, including DeLoRA, DoRA, and QR-LoRA, among others. Various formulations of norm or magnitude regularization have already been proposed to stabilize or constrain LoRA updates. Although the authors make an effort to highlight distinctions between their approach and some of these existing methods, the conceptual novelty appears incremental rather than fundamental.

2. Similar to previous works in this area, the practical application of selecting an appropriate norm bound remains challenging. In particular, it is often unclear how to determine suitable values for the target norm in practice. Moreover, the reviewer has concerns about whether enforcing a norm constraint is necessary or beneficial in low-rank settings (see my Question 2).

3. Incomplete Experimental Comparison: Given the close similarity of the proposed algorithm to existing norm-control methods, a more comprehensive experimental comparison is necessary. In particular, evaluations against representative baselines such as DeLoRA and QR-LoRA would provide a clearer assessment of the proposed method’s advantages. However, the current experiments primarily compare only with standard LoRA and DoRA, which limits the strength of the empirical evidence supporting the claimed improvements.

**Questions:**

1. Norm Choice in Practice: Could the authors elaborate on situations where nuclear, Frobenius, or spectral norm control produces meaningfully different outcomes, and provide further intuition regarding the practical selection of the $p$-norm in real setups? Moreover, how to set a proper norm value in these cases?

2. [Key Issue] The reviewer is concerned about whether controlling the norm of LoRA weights is necessary or appropriate in low-rank settings. In many cases, such as when the pre-trained model is suboptimal and certain weights behave randomly for downstream tasks, larger LoRA updates may be required to compensate for these deficiencies. Moreover, based on the reviewer’s hands-on experience, for very low ranks (e.g., rank-1), the learned LoRA matrices naturally exhibit large spectral norms, and constraining them could hinder expressiveness and degrade performance. This has also been observed in numerous recent studies, where singular values often tend to be extremely large after the LoRA training. For example, In figure 1 of [1], the authors show the singular values are amplified significanlty in order to learn some new features and dominate the original weight matrix.

3. Generality and Limitations of the Cayley-based Parameterization: Are there cases where the Cayley transformation could become numerically unstable, or where the mapping’s coverage is insufficient for adaptation, particularly for very large ranks or nearly non-square adaptation blocks?

4. The reviewer has concerns about the rationale presented around Line 123. Although the weights of the LoRA matrices may increase during training, the term $\partial L / \partial y$ typically decreases as the loss diminishes. As a result, the overall gradient magnitude may not necessarily grow excessively. Therefore, the explanation provided in this part does not seem fully convincing.

[1] Weight Spectra Induced Efficient Model Adaptation.

---

> ### Author Response · Authors · 2025-11-21
> **Part 1**
>
> We thank the reviewer for constructive comments. Here are our responses to your concerns and questions.
>
> ### **(W1) Novelty compared with existing approaches**
>
> We agree that several prior works have explored norm control for LoRA updates. However, our contribution is **not** another regularization strategy or post-hoc constraint. Instead, our work introduces a **theoretically grounded formulation** and a **complete reparameterization** that differ fundamentally from existing approaches. The key distinctions are summarized below.
>
> 1. *A Smooth and complete parameterization of all matrices within the prescribed norm and rank bounds.*
>
> Smoothness is necessary for stable optimization while completeness ensures strong expressive power, in particular when a tight bound is specified. As highlighted by Reviewer SKyQ, turning norm-bounded adaptation into a smooth and complete reparameterization is **novel and theoretically grounded**. Reviewer HAkH mentioned that our main theoretical result (Thm. 4.2) is **an original and elegant solution**. To the best of our knowledge, our parameterization is the **first** approach that covers all weight matrices satisfying arbitrarily prescribe bounds on all singular values. This  enables control over a variety of matrix norms beyond Frobenius norm, as noted by the reviewer.
>
> 2. *A principled mechanism for tight and predictable norm control*
>
> Most existing methods manage the norm **indirectly**:
>
> - DoRA [1] decomposes magnitude and direction but does not constrain singular values.
> - QR-LoRA [2] ensures orthogonality but does not directly control update magnitude.
> - DeLoRA [3] can enforce a Frobenius-norm bound, but good practical performance often requires a unconstrained learnable bound; this behavior is analyzed in detail in our response to Point (W3).
>
> In contrast, NB-LoRA provides **hard, explicit, and constant control** of the rank and norm budget throughout training. This differs from penalty-based methods, which only enforce norm constraints approximately and depend on optimization dynamics.
>
> 3. *Empirical robustness supported by theoretical insights*
>
> A central goal of robust PEFT—emphasized by Reviewer HAkH (S3)—is to maintain good performance while improving robustness to hyperparameters and preventing catastrophic forgetting. Our experiments across LLMs and ViTs show that NB-LoRA outperforms or matches existing methods across tasks and remains stable over a wide range of hyperparameters, especially the learning rate. Furthermore, our analysis on training dynamics provides theoretical insights on how NBLoRA produces stable updates, avoids small initial gradients, and prevents norm explosion, see Sec. 3–4 of the paper and our response to Reviewer SKyQ for more details.
>
> **References:**
>
> [1] Liu et al. DoRA: Weight-Decomposed Low-Rank Adaptation. ICML 2024.
>
> [2] Yang et al. QR-LoRA: Efficient and Disentangled Fine-tuning via QR Decomposition for Customized Generation. ICCV 2025.
>
> [3] Bini et al. DeLoRA: Decoupling Angles and Strength in Low-Rank Adaptation. ICLR 2025.
>
> ### **(W2, Q2) Regarding the key issue**
>
> Our experimental results indicate that LoRA can learn low-rank updates with large spectral norms (see **Fig. 1 in the supplementary material**), which aligns with the reviewer’s hands-on experience. We agree that if a task inherently requires a very low-rank but high–spectral-norm update, then vanilla LoRA may indeed be more suitable than NB-LoRA.
>
> However, our findings also show that this behavior substantially increases LoRA’s **sensitivity to the learning rate**. When the learning rate is moderately large, the training dynamics can become unstable; we provide a detailed illustration of this phenomenon in our response to Q4.
>
> In contrast, we argue that explicit norm control, as implemented in NB-LoRA, is essential when robustness is a priority. A core goal of our work is to develop a PEFT method that is stable across a broad range of hyperparameters while still achieving competitive performance—a point emphasized by Reviewer HAkH (S3). NB-LoRA achieves this by ensuring predictable update magnitudes, thereby offering a more robust alternative in scenarios where robustness is critical.

---

> ### Author Response · Authors · 2025-11-21
> **Part 2**
>
> ### **(W3) Comparison experiments on DeLoRA and NBLoRA**
>
> First, we would like to clarify why DeLoRA was not included as a baseline in Table 2. First, as shown in **Fig. 3 (see the supplementary material)**, DeLoRA typically requires much larger learning rates than the baselines reported in Table 2 to reach comparable performance, e.g., [5e-3, 5e-2] v.s. [5e-5, 5e-4]. Second, because DeLoRA also provides Frobenius-norm control, it is much closer to our method than the other baselines. For this reason, we believe a direct comparison between DeLoRA and NB-LoRA is more meaningful; an initial comparison was provided in Fig. 4 of the paper. Following Reviewer SKyQ’s comments, we expanded this analysis by studying their training dynamics in more detail.
>
> DeLoRA can be written as
> $$
> W_{\mathrm{delora}}=\frac{\delta}{2r}B^\top \Xi A- \frac{\delta_0}{2r}B_0^\top \Xi_0 A_0
> $$
> with $\Xi = \mathrm{diag}\bigl(1/(|a_{i}|\cdot |b_i|)\bigr)$ where $a_i,b_i$ are the $i$th rows of $A,B$, respectively. When $\delta$ is fixed to be $\delta_0$, this method guarantees a Frobenius norm bound of $\delta_0$, which we denote as *DeLoRA*$_{\delta=\delta_0}$. If $\delta$ is instead learned, this guarantee no longer holds; we denote this variant simply as *DeLoRA*.
>
> Following the same training-dynamics derivation used for NB-LoRA and PiSSA, DeLoRA can be rewritten as
> $$
> W_{\mathrm{delora}}=\hat{B}^\top \hat{A} - \hat{B}_0^\top \hat{A}_0 \quad \text{with}\quad \hat{A}=\sqrt{\frac{\delta}{2r}}\mathrm{diag}\left(\frac{1}{\|a_i\|_2}\right)A \text{ and } \hat{B}=\sqrt{\frac{\delta}{2r}}\mathrm{diag}\left(\frac{1}{\|b_i\|_2}\right)B,
> $$
> from which we obtain the constraint
> $$
> \\|\hat{A}\\|_F^2+\\|\hat{B}\\|_F^2=\delta.
> $$
> NB-LoRA satisfies an analogous relation:
> $$
> \\|\hat{A}\\|_F^2+\\|\hat{B}\\|_F^2=2\delta\mathrm{trace}(\tilde{S})\approx 2\delta \sqrt{r}
> $$
> since $\tilde{S}\approx\frac{1}{\sqrt{r}}I$ when Frobenius norm is considered. Here the R.H.S. of NB-LoRA is $2\sqrt{r}$ times as that of DeLoRA. Thus, NB-LoRA permits much larger $\\|\hat{A}\\|_F$ and $\\|\hat{B}\\|_F$ while maintaining the same global norm bound as DeLoRA with fixed $\delta$. This means that NB-LoRA is more expressive, a point also illustrated geometrically in Fig. 2 of the paper. Our analysis above indicates the the expressive power difference grows as the rank $r$ increases.
>
> A second implication concerns the learning speed. The updates satisfy
> $$
> \Delta \hat{A} \approx -\eta \frac{\partial \ell}{\partial \hat{A}}= -\eta \hat{B} D_{xy}, \quad \Delta \hat{B}\approx -\eta \frac{\partial \ell}{\partial \hat{B}}= -\eta \hat{A} D_{xy}^\top,\quad \Delta W\approx \hat{B}^\top (\Delta \hat{A})+(\Delta \hat{B})^\top \hat{A}
> $$
> where $D_{xy}=(\partial \ell/\partial y)x^\top$ and $\eta$ is the learning rate. Since NB-LoRA has much larger $\hat{A}$ and $\hat{B}$ than DeLoRA, it is expected that NB-LoRA will also yield large parameter update. As shown in **Fig. 3** (see the supplementary material), NB-LoRA achieves substantially **larger parameter updates** $(\Delta\hat{A},\Delta\hat{B},\Delta W)$ than DeLoRA—even though NB-LoRA uses a **smaller learning rate** (1e-3 vs. 5e-3). Making $\delta$ learnable for DeLoRA alleviates this limitation to some extent, but it eliminates the norm-bound guarantee, and the GSM8K accuracy of DeLoRA is still lower than NB-LoRA.
>
> ### **(Q1) Norm choice in practice**
>
> **(Choice of norm type).** A practical scenario where the choice of norm matters is federated fine-tuning [4]. When aggregating weight updates $W$ from multiple nodes under differential privacy constraints, enforcing a Frobenius-norm bound per node is a natural choice. However, in heterogeneous data settings, a spectral-norm bound can provide more robust merging and yield more stable global updates. NB-LoRA accommodates both types of constraints through the parameterization of singular-value bound $S$, allowing practitioners to choose the norm most appropriate for their setup.
>
> **(Choice of norm bound).** As discussed in our response to Reviewer xuPe, we performed extensive hyperparameter sweeps over the bound $\delta$. Both our empirical findings and theoretical analysis indicate that $\delta$ behaves similarly to a regularization coefficient, for which simple and predictable tuning strategies exist. In particular, for LLM experiments in this paper, initializing $\delta$ such that the induced scaling factor is close to 1 consistently yields good performance and serves as a default.
>
> [4] Wang et al. FLoRA: Federated Fine-Tuning Large Language Models with Heterogeneous Low-Rank Adaptations. NeurIPS 2025.

---

> ### Author Response · Authors · 2025-11-21
> **Part 3**
>
> ### **(Q3) Limitations on Cayley-based parameterization**
>
> Our theoretical result (Thm. 4.2) establishes that the proposed parameterization covers **all** matrices $W\in \mathbb{R}^{m\times n}$ whose $i$th singular value is bounded by $s_i$ with
> $$
> s_1\geq s_2\geq \cdots \geq s_r> s_{r+1}=0
> $$
> where $r\leq \min(m,n)$ is the prescribed rank budget. With the additional parameterization of the vector $s$ (Lines 190 - 200), NB-LoRA achieves complete coverage of all matrices whose rank and Schatten $p$-norm are simultaneously bounded. This guarantee holds for both square and non-square matrices and for any feasible rank $r$. The completeness proof (Appendix A–B) is constructive: given any admissible $W$, one can follow Eqs. (12)–(15) and (8)–(9) to obtain free parameters $(\tilde{A},\tilde{B})$ such that $W=\mathcal{W}_S(\tilde{A}, \tilde{B})$ where $\mathcal{W}_S$ is defined in Eq. (3) of the paper.
>
> The main practical limitation of NB-LoRA lies in the computational overhead for very large ranks. As shown in Table 3, the cost remains comparable to LoRA up to $r=256$. Fortunately, most LoRA applications use $r\leq 128$, where NB-LoRA remains comparable as LoRA.
>
> Since $\tilde{A},\tilde{B}$ are free learnable parameters, by further increasing the learning rate and norm bound, we can observe instability in training dynamics. In practice, however, our experiments show that NB-LoRA remains stable across a wide range of learning rates.
>
> ### **(Q4) About rational around Line 123**
>
> Here we will elaborate the rational around Line 123 based on the training dynamics of LoRA (see **Fig. 4 in the supplementary material**). We agree with the reviewer that although the norm of $W_{\mathrm{lora}}$ increases during training, it can remain bounded if the term $\frac{\partial \ell}{\partial y}$ decreases sufficiently. This behavior is indeed observed under small learning rates, as shown in **Fig. 4a**.  However, this is not guaranteed for larger learning rates. As shown in **Fig. 4b**, when lr=1e-3, the norms of $W_{\mathrm{lora}}$ grow rapidly.
>
> In particular, as shown in **Fig. 4** (col. 4), the difference between the Frobenius and spectral norms is small, suggesting that LoRA learns a single dominating singular value. At Step 40, we observe an abrupt increase in the gradient norm, while the spectral norm remains smooth. Based on the following gradient formulas
> $$
> \frac{\partial \ell}{\partial A}=\frac{\alpha}{r}B \left(\frac{\partial \ell}{\partial y}\right)x^\top ,\quad \frac{\partial \ell}{\partial B}=\frac{\alpha}{r}A x\left(\frac{\partial \ell}{\partial y}\right)^\top,
> $$
> we can conclude this large gradient spike is due to the term $\partial \ell/\partial y$, as the input $x$ is typically bounded due to the normalization layer. This is further leads to the sharp increase in the loss curve observed around Step 45.
>
> As discussed in our response to Reviewer SKyQ, NBLoRA provides tighter norm control and distributes updates across multiple directions. In **Fig. 1b** (see the supplementary material), under a large learning rate, the nuclear norm of $W_{\mathrm{nblora}}$ approaches its prescribed bound of $\delta=128$, while its spectral norm remains small, which helps to improve training stability.

---

> ### Comment · Reviewer_smpu · 2025-11-27
> **Response to Authors' Rebuttal**
>
> First of all, the reviewer would like to thank the authors for their detailed response.
>
> Regarding my key concern, I do not believe the current rebuttal adequately addresses the issue. The authors argue that LoRA is highly sensitive to the learning rate and, on that basis, emphasize the necessity of enforcing a bounded norm. However, this justification is not fully convincing. If learning-rate sensitivity were the core problem, there exist simpler and more economical remedies—for example, using a smaller learning rate or employing a more aggressive decay schedule. These alternatives do not suggest a fundamental limitation of LoRA itself.
>
> Moreover, the reviewer remains unconvinced that strictly constraining the norm is an ultimately desirable solution. Counterexamples are easy to construct, and a bound may introduce its own optimization difficulties. Approaches based on adaptive or dynamic norms may provide a more flexible and theoretically grounded alternative.

---

> > ### Author Response · Authors · 2025-12-04
> >
> > Thank you for the constructive discussion. We agree with the reviewer that there exist fine-tuning tasks that *do* require high–spectral-norm updates in order to encode new knowledge from the downstream dataset. For such tasks, vanilla LoRA may indeed be more suitable than NB-LoRA. We have added this clarification to the limitations section in the revised version.
> >
> > To address the reviewer’s concerns regarding the practical usefulness of norm control, we have added new DreamBooth fine-tuning experiments. This task lies in the low-data regime, with each subject containing only 5–6 images. We show that LoRA and its variants (DoRA, PiSSA) suffer from overfitting under this setting. In contrast, due to its tighter norm constraint, our method effectively mitigates overfitting and produces generated images with improved prompt fidelity and sample diversity.

---

### Official Review · Reviewer_HAkH · 2025-10-19

**Soundness:** 3
**Presentation:** 3
**Contribution:** 3
**Rating:** 6
**Confidence:** 4

**Summary:**

This paper proposes a reparametrization of LoRA via Cayleigh transform that enables norm-bounded fine-tuning, i.e. the learned weight delta is bounded in its distance to the pretrained weights as measured by Schatten-Norm. The paper then shows that this method has similar or better fine-tuning performance compared to baseline methods (Tab. 2 and Fig. 6), while being more robust to learning rate (Fig. 3) and mitigating catastrophic forgetting (Fig. 5). Finally, it analyzes the computational overhead of the proposed method, finding somewhat increased computational demands over baselines, which can be mitigated by using a computational setup for the backward pass through the Cayley transform especially designed for this setup (Appendix C).

**Strengths:**

**(S1)** The main insight (Thm. 4.2) is original and non-obvious. I find this an elegant solution.

**(S2)** Discussion of related work and positioning relative to it is excellent. The main paper in detail discusses similarities and differences to DeLoRA and PiSSA, including detailed experiments on the attainable norm bounds (Fig. 4b) and which matrices can be learned (Fig. 2).

**(S3)** The experiments are on point, to me the main evaluation of robust peft methods is not that they surpass the performance of existing methods, but roughly match their performance while retaining strengths of the original model (i.e. no catastrophic forgetting) and being less sensitive to hyperparameters, especially the learning rate. Both domains are covered in this paper and the results convincingly demonstrate the merits of the proposed method.

**(S4)** Describing the custom backward pass for the Cayleigh transform in Appendix C is a great addition to the paper and significantly increases its practical value.

**(S5)** Computational overhead incurred by the proposed method is sufficiently discussed.

**(S6)** Design decisions are generally well justified, for example, line 197 (unfortunately this equation does not have label)

**(S7)** The motivation in Sec. 3 is helpful and interesting to understand the merits of norm-bounded peft approaches.

**Weaknesses:**

**(W1)** Experiments only consider the high-data regime (MetaMathQA=395K samples, CodeFeedback=66.5K samples). However, to test the robustness of models, it would also be interesting to see if the proposed method makes models more robust to overfitting in low-data regime, for example, through the Dreambooth task used in DeLoRA or similar.

**(W2)** Appendices E and F mention that experiments use lr schedulers (cosine for LLMs, one-cycle for ViTs). This is unfortunate because it obscures the effect of scheduler vs. norm-bound on learning-rate robustness. Consider adding an ablation that demonstrates robustness with a constant learning rate, and ideally different schedulers, if compute budget allows.

**(W3)** I couldn't find the results for prolonged training mentioned in Appendix E (line 847). Perhaps they are in the paper, but the description is not clear.

**(W4)**  The paper convincingly shows _that_ the method works. However, is there any intuition as to _why_ the reparametrization in Eq. 3 works? I think this could help readers understand the method better and also help enable further research building on top of the proposed method.

### Minor Weaknesses:
  *  Line 18: "avoid model catastrophic forgetting without minor cost on adaptation performance": This sentence is (semantically) unclear to me.
   * References: Overall, please check carefully in which cases `\citep` is appropriate, and when `\citet`. I see many instances where this is not intuitive.
   * There are some typos and grammatical errors, for example, line 166.
   * Some equations don't have labels (mainly on page 4), but it would be convenient to have them to be able to refer to them, e.g. from other papers
   * I find it confusing that captions appear below figures, but above tables. Consider placing captions below tables as well

**Questions:**

* Does the proposed method help prevent overfitting in low-data regime finetuning?
* What are the separate effects of learning rate scheduler and NB-LoRA on learning rate robustness?
* Is there any intuition that can be compactly explained in natural language, why the proposed reparameterization works?

I am already leaning toward the Rating "8 (accept)". I hope these points and other mentioned weaknesses can be addressed in the rebuttal, and I am looking forward to a constructive discussion with the authors.

---

> ### Author Response · Authors · 2025-11-21
>
> We thank the reviewer for careful reading and positive comments. Here are responses to the comments and suggestions.
>
> ### **(W1, Q1) Experiments in low-data regime**
>
> We agree that evaluating robustness in low-data settings (e.g., the DreamBooth task used in DeLoRA) is an interesting and important direction for our approach. Our comparison with DeLoRA on NLG tasks shows that both methods can tightly control the Frobenius norm of the weight update. This suggests that NB-LoRA may also exhibit similar robustness in low-data scenarios such as DreamBooth. However, due to time constraints, we were not able to include such experiments in the current version. We view this as a promising direction for future work.
>
> ### **(W2, Q2) Robustness under constant learning-rate scheduler**
>
> Following the reviewer’s suggestion, we conducted an ablation study using a constant LR scheduler, see the tables below. Across a wide range of learning rates, NB-LoRA remains significantly more robust than both LoRA and PiSSA. Compared with the cosine scheduler, NB-LoRA’s average GSM8K accuracy decreases by only **3.2\%**, whereas PiSSA decreases by 14.3\%. For LoRA, training diverges at even moderate learning rates (5e-4).
>
> - Constant LR scheduler, GSM8K accuracy
>
> |LR|5e-5|1e-4|5e-4|1e-3|Avg|
> |:-:|:-:|:-:|:-:|:-:|:-:|
> |LoRA|50.1|55.6|failed|failed|- |
> |PiSSA|54.0|54.4|37.8|18.5|41.2|
> |NBLoRA|56.7|56.6|51.5|52.2|54.3|
>
> - Cosine LR scheduler, GSM8K accuracy
>
> |LR|5e-5|1e-4|5e-4|1e-3|Avg|
> |:-:|:-:|:-:|:-:|:-:|:-:|
> |LoRA|47.6|52.2|58.3|failed|- |
> |PiSSA|56.3|60.6|55.8|49.4|55.5|
> |NBLoRA|53.4|57.8|59.7|59.2|57.5|
>
> ### **(W3) Prolonged training results**
>
> Thank you for noting this. The prolonged-training results do appear in Figure 8 of the paper, but we mistakenly did not cite them explicitly in the relevant discussion. We apologize for this confusion. In the revised version, we will update the text to reference the figure and provide a clearer explanation of the findings.
>
> ### **(W4, Q3) Is there any compact language to explain why it works**
>
> The main features of the proposed approach are summarized below:
>
> The NB-LoRA parameterization has builtin guarantee of a structural constraint on the LoRA weights that simultaneously avoid small initial gradients, prevent uncontrolled growth of the adaptation norm, and maintain well-conditioned updates throughout training. These properties together explain its strong robustness and good empirical performance across our experiments.
>
> ### **Minor points**
>
> Thank you for identifying these issues. We will address all of them in the revised version.

---

> ### Comment · Reviewer_HAkH · 2025-11-21
>
> Thank you for the detailed answer to my review.
>
> ---
>
> I noticed that authors did not yet upload a rebuttal revision, and also referred to changes in future tense. Authors may consider these instructions from the [Author Guide](https://iclr.cc/Conferences/2026/AuthorGuide):
> > During the discussion/rebuttal phase and for the camera ready, the page limit will be increased to 10 pages to allow for new results/discussions.
>
> This allows to update the paper and increase its length to 10 pages. Usually, changes compared to the original submission are highlighted in blue in the revision. I recommend that authors make use of this opportunity.
>
> ---
>
> > (W1, Q1) Experiments in low-data regime
>
> Thank you for the answer. Indeed, I think this experiment would be a strong addition to the paper, and I agree with the authors that the proposed method is likely to be helpful because it allows for controlling the norm of the update.
>
> On a related note, I think this experiment could help address concerns of other reviewers (for example, Reviewer smpu and xuPe). At least in my understanding and experience, one of the main motivations for controlling the update strength is to prevent overfitting, which typically occurs in low-data regimes.
>
> ---
>
> > (W2, Q2) Robustness under constant learning-rate scheduler
>
> I acknowledge that these experiments clarify my doubts. Please do include them in the revision.
>
> ---
>
> > Prolonged training results
>
> Thank you for the clarification.
>
> ---
>
> > (W4, Q3) Is there any compact language to explain why it works
>
> The intention of my question was to give an explanation of how the components of the proposed method lead to the described advantages. In my view, this can be expanded upon, and the description given in the rebuttal mainly lists the advantages. Please consider expanding this in the revision.
>
> ---
>
> As my concern regarding effect of learning rate scheduler is resolved and I am convinced the proposed method will help in low data regime as well even if authors do not show experiments, I think the paper is an interesting contribution and should be presented at the conference, according to my assessment.
>
> I will update my initial rating to 8 once authors have provided the rebuttal revision and I have confirmed that minor changes and experiments have been included.

---

> > ### Author Response · Authors · 2025-11-22
> >
> > We thank the reviewer for the prompt follow-up discussion and the positive evaluation of our work.
> >
> > - **Rebuttal revision and low-data-regime experiment**
> >
> > Thank you for your feedback. We will begin preparing the revised manuscript and work on the DreamBooth task as suggested.

---

> > > ### Author Response · Authors · 2025-12-04
> > >
> > > We thank the reviewer for the very helpful suggestion. We have added DreamBooth experiments in the revised version. As the reviewer anticipated, tighter norm control indeed helps prevent overfitting in the low-data regime.

---

### Official Review · Reviewer_SKyQ · 2025-10-26

**Soundness:** 2
**Presentation:** 3
**Contribution:** 3
**Rating:** 4
**Confidence:** 3

**Summary:**

This paper proposes a norm-bounded reparameterization for LoRA that directly controls the magnitude of low-rank updates during fine-tuning, aiming to improve training stability and reduce forgetting. The key idea is to turn a constrained optimization (on the Schatten/Frobenius/spectral norm of $\Delta W$) into an unconstrained learning problem via a smooth reparameterization, backed by theoretical guarantees. Experiments indicate solid downstream performance and less forgetting across settings.

**Strengths:**

- Turning norm-constrained low-rank adaptation into a smooth, complete reparameterization is novel and comes with clear theoretical underpinnings.
- The paper presents experiments in both language and vision models suggesting both performance gains and reduced forgetting, accompanied by useful analyses of training dynamics

**Weaknesses:**

- Gradient-dynamics mismatch in analyses/plots: The method trains *free* parameters $\tilde A,\tilde B$ that map to $A,B$ through a Cayley transformation. Under a given learning rate, the *effective* update on $A,B$ is not simply $-\eta\nabla_{A,B}L$; it is mediated by the Jacobian of the reparameterization. Therefore, comparing raw gradient norms (or update magnitudes) across methods in different parameterizations can be misleading. A fair comparison should report the induced per-step update on $A,B$ under the same learning rate (e.g., by mapping the optimizer step in $\tilde A,\tilde B$ to the resulting $\Delta A,\Delta B$. This would align the empirical evidence with the paper’s intuition that the reparameterization accelerates/steadies optimization.
- Forgetting claim not causally tied to “smaller updates”: The motivation argues that bounding the parameter update magnitude should mitigate forgetting, but the experiments do not directly validate this causal link. The paper should (i) quantify the post-training update magnitude of $\Delta W$ across methods under matched budgets, (ii) correlate these magnitudes with the measured forgetting, and (iii) show that the proposed norm bound achieves the claimed trade-off because of smaller/effectively shaped updates rather than unrelated side effects.

**Questions:**

- Could the authors explain more on how to pick the norm bound $\delta$ in practice? In the ViT experimental results, it seems that the performance is insensitivity within a range. However, we can imagin that extremely small $\delta$ should over-constrain adaptation while very large $\delta$ could negate the benefits claimed in the paper. Could the authors provide actionable guidance on estimating a suitabel range for $\delta$?

---

> ### Author Response · Authors · 2025-11-21
> **Part 1**
>
> We thank the reviewer for careful reading and very helpful comments.
>
> ### **(W1) Training dynamics analysis**
>
> We agree with the reviewer that comparing raw gradient norms across methods is not appropriate due to different parameterizations. Thank you for highlighting this issue and for the very constructive suggestions.
>
> Following your comments, we have replaced the raw gradient-norm plots with norms of the weight matrices and their increments, see **Fig. 1 in the supplementary material**. To enable comparison across LoRA, PiSSA, and NB-LoRA, we first express all methods in a unified form. Specifically, we rewrite:
>
> - LoRA as $W_{\mathrm{lora}}=\hat{B}^\top \hat{A}$ with $\hat{A}=\sqrt{\alpha/r}A$ and $\hat{B}=\sqrt{\alpha/r}B$;
> - PiSSA as $W_{\mathrm{pissa}}=\hat{B}^\top \hat{A}-\hat{B}_0^\top \hat{A}_0$;
> - NBLoRA as $W_{\mathrm{nblora}}=\hat{B}^\top \hat{A}$ with $\hat{A}=\sqrt{2\delta}\tilde{S}^{1/2}A$ and $\hat{B}=\sqrt{2\delta}\tilde{S}^{1/2}B$.
>
> Under this representation, we have
> $$
> \Delta \hat{A} \approx -\eta \frac{\partial \ell}{\partial \hat{A}}= -\eta \hat{B} D_{xy}, \quad \Delta \hat{B}\approx -\eta \frac{\partial \ell}{\partial \hat{B}}= -\eta \hat{A} D_{xy}^\top,\quad \Delta W\approx \hat{B}^\top (\Delta \hat{A})+(\Delta \hat{B})^\top \hat{A}
> $$
> where $D_{xy}=(\partial \ell/\partial y)x^\top$ and $\eta$ is the learning rate. As mentioned in our response to Reviewer xuPe, we choose the hyperparameters $\alpha$ and $\delta$ such that all methods have the same effective scaling factor. Below we summarize the training behavior of each method and relate our findings to the intuitions in Sections 3 and 4 of the paper.
>
> **(LoRA).** Because $\hat{A}$ is initialized as a small random matrix and $\hat{B}$ initialized as zero,  LoRA exhibits very small updates $\Delta W$ for an extended period when the learning rate is small, see **Fig. 1a in the supplementary material**.  Large learning rate can help to mitigate this issue during the initial training stage, see **Fig. 1b**. A larger learning rate alleviates this issue during the early phase, but may cause training instability. In **Fig. 1 (col. 4)**, LoRA shows a relatively small nuclear norm but a much larger spectral norm, indicating that the updates tend to concentrate on a very low-rank subspace, which leads to instability for a large learning rates (1e-3).
>
> **(PiSSA).** Different from LoRA, PiSSA initializes $\hat{A}$ and $\hat{B}$ based on dominant singular components of the pretrained weights [1], leading to significantly larger updates even when the learning rate is small. However, without explicit norm control, the norm of $W_{\mathrm{pissa}}$ increases substantially for large learning rates (**Fig. 1b**), sometimes overwriting useful pretrained structure. This explains its sensitivity to the learning rate observed in Fig. 3 and Table 2 of the paper.
>
> The fourth column of **Fig. 1** shows that PiSSA tends to have larger nuclear norm but smaller spectral norm than LoRA, suggesting a more uniform singular value distribution. This makes PiSSA more stable than LoRA at high learning rates, but potentially harmful for performance when the update magnitude becomes too large.
>
> **(NBLoRA).** Since $A,B$ are parameterized via Cayley transformation, we have $AA^\top+ BB^\top=I$ and
> $$
> \hat{A}\hat{A}^\top + \hat{B}\hat{B}=2\delta \tilde{S}^{1/2}(AA^\top+BB^\top)\tilde{S}^{1/2}=2\delta\tilde{S},
> $$
> which implies $\\|\hat{A}\\|_F^2+\\|\hat{B}\\|_F^2=2\delta \cdot \mathrm{trace}(\tilde{S})$. Thus, the combined parameter matrix $X:=\begin{bmatrix}
>     \hat{A} & \hat{B}
> \end{bmatrix}$ lies on a compact manifold which is closely related the Stiefel manifold. Empirically (**Fig. 1**, col. 3, see the supplementary material), we can verify that $\\|\hat{A}\\|_F^2+\\|\hat{B}\\|_F^2\approx 2\delta=256$ as $\tilde{S}\approx \frac{1}{r}I$ and $\delta=r=128$. Hence, $\hat{A},\hat{B}$ cannot be both small matrices, which address the small initial gradient issue in the LoRA parameterization. As shown in **Fig. 1** (col. 2), NBLoRA exhibits larger updates $\\|\Delta \hat{A}\\|_F$ and $\\|\Delta \hat{B}\\|_F$ than LoRA and PiSSA. On the other hand, because increasing $\\|\hat{B}\\|_F$ will also deceases $\\|\hat{A}\\|_F$ simultaneously, the norm of $W$ remains tightly controlled. With a larger learning rate, NB-LoRA attains the active bound while maintaining stability due to its more uniform singular-value distribution.
>
> [1] Meng et al. PiSSA: Principle singular values and singular vectors adaptation of large language models. NeurIPS 2024.

---

> > ### Author Response · Authors · 2025-11-21
> > **Part 2**
> >
> > ### **(W2) The link between forgetting and update magnitude**
> >
> > Thank you for your suggestions. We now include plots of the **norm ratio** $\\|W\\|/\\|W_{\mathrm{pretrain}}\\|$ (maximized over all PEFT blocks) alongside the performance metric, see **Fig. 2 in the supplementary material**. As seen in the top left of **Fig. 2**, when the norm budget $\delta$ of NB-LoRA increases, the corresponding adapter norm also increases and the model forgets more on the source task (ImageNet-1k). Similar trends are observed in other adapters. However, across methods, this relationship does not necessarily hold—for example, VeRA may have larger norms but less forgetting than AdaLoRA. Due to the ability of tight norm control, NB-LoRA consistently exhibits substantially less forgetting than other methods while maintaining good adaptation performance.
> >
> >
> > ### **(Q1) How to choose hyper-parameter $\delta$**
> >
> > We performed hyperparameter searches for the LLM experiments (see response to Reviewer xuPe). Both empirical evidence and theory show that the norm bound $\delta$ behaves like a **regularization coefficient**. A practical heuristic is to initialize $\delta$ such that NB-LoRA has an effective scaling factor of 1, and then tune $\delta$ using some rules for regularization hyperparameters.

---

### Official Review · Reviewer_xuPe · 2025-10-28

**Soundness:** 3
**Presentation:** 3
**Contribution:** 3
**Rating:** 4
**Confidence:** 4

**Summary:**

The paper proposes Norm-Bounded Low-Rank Adaptation (NB-LoRA), a PEFT method that enforces explicit norm constraints on low-rank weight updates via a smooth, complete reparameterization based on the Cayley transform. This formulation ensures stability during optimization by bounding singular values under any unitarily invariant norm. Experiments on large language models and vision transformers show that NB-LoRA achieves comparable or superior performance to existing PEFT methods while maintaining strong robustness to hyperparameters. The approach introduces minimal computational overhead and offers a principled framework for norm-controlled model adaptation.

**Strengths:**

1. **Clear motivation.** The paper targets a well-documented limitation of standard LoRA—small initial gradients and sensitivity to learning rate—and proposes a theoretically grounded reparameterization that tightly controls singular values and associated norms. The mapping is smooth and complete, covering all matrices within the prescribed rank and singular-value bounds
2. **Thorough empirical study.** The evaluation spans both LLMs and ViTs, includes comparisons against multiple baselines, ablations, and analyses of training dynamics, norm saturation, and hyperparameter robustness. The large-model case (LLaMA-3-70B) is also reported with memory and time.

**Weaknesses:**

1. **Additional Hyperparameters**: NB-LoRA introduces additional control (e.g., the norm bound $\delta$). Although the paper argues for improved robustness, this still expands the tuning surface in practice. And how to balance the performance and robustness by tuning $\delta$ is also a trade-off.
2. **Extra training time / GPU overhead.** The Cayley reparameterization (and its backward pass) adds measurable overhead relative to vanilla LoRA. But it's minor according to the paper.

**Questions:**

1. **Important**: Using the code from the Supplementary Material, I observe a higher training time when the micro-batch size is 1. The measured per-step times and memory are:

|         | Micro Batch Size | Time (per step) | GPU memory |
| ------- | ---------------- | --------------- | ---------- |
| LoRA    | 1                | 4.80            | 18643MiB   |
| NB-LoRA | 1                | 8.13            | 18859MiB   |
| LoRA    | 8                | 3.81            | 55309MiB   |
| NB-LoRA | 8                | 3.95            | 55601MiB   |
 I just modified my codebase according to the provided code. Specifically, in vanilla LoRA implementation, the core forward pass is:
   ```python
   result = result + lora_B(lora_A(dropout(x))) * scaling
   ```
I copied  ``Cayley(torch.autograd.Function)`` in ``code/llm/peft/tuners/lora/layer.py`` to my codebase and changed the forward pass to :
```python
   # result = result + lora_B(lora_A(dropout(x))) * scaling
   GH = Cayley.apply(torch.cat([lora_A.weight.T, lora_B.weight], dim=0))
   A, B = GH[:self.in_features, :], GH[self.in_features:, :].T
   result = result + (((dropout(x) @ A) * 1) @ B) * 2.0
```
Could the authors explain why the **time per step nearly doubles** at micro-batch size 1 while it becomes comparable at micro-batch size 8? Is this due to the cost of constructing G,HG,HG,H (including the r×rr\times rr×r inverse and custom backward), the additional autograd graph, or the loss of kernel fusion compared to standard LoRA?

**Experiment setup:** Model = LLaMA-2-7B-HF; Dataset = MetaMathQA; Global batch size = 32; Micro-batch size = 1.

2. In Table 1 (LLaMA-3-70B on GSM8K), why PiSSA appears to underperform LoRA at the smallest learning rate?
3. What is the difference of Figure 4a and Figure 3a for NB-LoRA？I notice that in Figure 4a NB-LoRA has a higher test accuracy.
4. The Cayley transform in Eq4 yields a semi-orthogonal matrix $G\in R^{(m+n) \times r }$. Splitting $G$ into blocks gives $A^T \in  R^{n \times r}$ and $B^T \in  R^{m \times r }$. However, this procedure may produce A and B that are not themselves orthogonal and may also introduce an imbalance between A and B, which appears to conflict with the discussion in the Motivating Analysis section.”

**Additional notes to the authors**: Your Hugging Face API key appears to be exposed in the uploaded supplementary material.

---

> ### Author Response · Authors · 2025-11-21
> **Part 1**
>
> We thank the reviewer for the thoughtful and constructive feedback. We address each point below.
>
> ### **(W1) Additional Hyperparameter**
>
> We want to clarify that NB-LoRA does **not** introduce any additional hyperparameters beyond those in standard LoRA, provided the norm type is specified. To be specific, standard LoRA uses two hyperparameters—scaling $\alpha$ and rank $r$—in the parameterization
> $$
> W_{\mathrm{lora}}=\frac{\alpha}{r} B^\top A
> $$
> The proposed NB-LoRa also involves two hyperparameters (bound $\delta$ and rank $r$). Taking the nuclear norm control as an example, the parameterization has the form of
> $$
> W_{nblora}=2\delta B^\top \tilde{S} A
> $$
> where $\tilde{S}$ is a diagonal matrix initialized close to $\frac{1}{r}I$. In this case, the effective scaling factor is $\frac{\delta}{r}$. The constant factor 2 is part of our parameterization (see Eq. (3) of the paper) and is not counted as part of the scaling. In the paper we use $\delta=\alpha$, ensuring that *LoRA and NB-LoRA used the same effective scaling*.
>
> Similarly, the effective scaling factor for NBLoRA with Frobenius and spectral norms are $\delta/\sqrt{r}$ and $\delta$, respectively. Throughout the LLM experiments in our paper, we choose $\delta$ based on $\alpha$ and $r$ so that both methods use an identical effective scaling factor:
> $$
> \delta=\begin{cases}
>     \alpha & \text{nuclear norm} \\\\
>     \alpha/\sqrt{r} & \text{Frobeius norm} \\\\
>     \alpha/r & \text{spectral norm}
> \end{cases}.
> $$
> Following prior work [1], we use $\alpha=r$ as the default in the initial submission.
>
> We additionally performed some hyperparameter search experiments. The following tables report results under different choices of $\alpha$ and the corresponding $\delta$. Note that these two hyperparameters are directly linked by the above formulas and therefore NBLoRA do **not** expand the tuning surface in practice.
>
> - LLaMA-2-7B, lr=5e-4, $r=128$
>
> |Scaling $\alpha$|16|32|64|128|256|
> |:-:|:-:|:-:|:-:|:-:|:-:|
> |LoRA|54.7|57.9|58.3|58.3|**60.3**|
> |Nuclear norm bound $\delta$|16|32|64|128|256|
> |NBLoRA|55.8|58.2|**60.6**|59.7|58.8|
> |Frobenius norm bound $\delta$|$\sqrt{2}$|$2\sqrt{2}$|$4\sqrt{2}$|$8\sqrt{2}$|$16\sqrt{2}$|
> |NBLoRA|56.0|57.6|58.5|58.8|**59.2**|
> |Spectral norm bound $\delta$|$\frac{1}{8}$|$\frac{1}{4}$|$\frac{1}{2}$|1|2|
> |NBLoRA|55.8|58.2|**61.2**|60.6|58.4|
>
> - LLaMA-2-7B, lr=1e-4, $r=128$
>
> |Scaling $\alpha$|16|32|64|128|256|
> |:-:|:-:|:-:|:-:|:-:|:-:|
> |LoRA|46.8|48.7|52.2|54.0|**57.0**|
> |Nuclear norm bound $\delta$|16|32|64|128|256|
> |NBLoRA|51.4|54.7|57.8|**60.1**|59.4|
>
> - Mistral-7b-v0.1, lr=5e-5, $r=128$
>
> |Scaling $\alpha$|16|32|64|128|256|
> |:-:|:-:|:-:|:-:|:-:|:-:|
> |LoRA|68.5|70.7|71.5|73.2|**73.8**|
> |Nuclear norm bound $\delta$|16|32|64|128|256|
> |NBLoRA|71.7|73.1|**74.7**|74.1|71.8|
>
> - Mistral-7b-v0.1, lr=5e-5, $r=32$
>
> |Scaling $\alpha$|8|16|32|64|128|
> |:-:|:-:|:-:|:-:|:-:|:-:|
> |LoRA|68.8|68.8|69.4|72.6|**72.8**|
> |Nuclear norm bound $\delta$|8|16|32|64|128|
> |NBLoRA|71.0|71.8|73.1|**73.9**|71.0|
>
> Across all settings, we observe two consistent trends. First, NB-LoRA achieves higher average and minimum performance as the hyperparameter varies. Second, the empirical results show that
> $\delta$ acts similarly to a regularization coefficient. That is, increasing $\delta$ from a small value will improves performance until a threshold, after which large $\delta$ leads to degradation.
>
> Here we give a theoretical explanation for this observation. NB-LoRA reparameterizes the following Ivanov regularization problem:
> $$
> \min_{W}\quad \ell(W)\quad \mathrm{s.t.}\quad \\|W\\|\leq \delta.
> $$
> which is closely related to the Tikhonov formulation:
> $$
> \min_{W}\quad \ell(W)+\lambda \\|W\\|.
> $$
> Under mild assumptions, these two problems are **equivalent** [2,3]. Since NB-LoRA provides a complete parameterization over the feasible set, the constrained problem can be expressed as an unconstrained one without loss of expressivity. Thus,
> $\delta$ has a similar effect to $\lambda$, explaining the empirical behavior observed above.
>
> **References:**
>
> [1] Meng et al. PiSSA: Principle singular values and singular vectors adaptation of large language models. NeurIPS 2024.
>
> [2] Oneto et. al. Tikhonov, Ivanov and Morozov regularization for support vector machine.  Machine Learning, 2016.
>
> [3] P. Goyal. Quora QA [link](https://www.quora.com/Is-there-a-connection-between-Tikhonov-and-Ivanov-Regularization-in-Machine-Learning), 2022.

---

> ### Author Response · Authors · 2025-11-21
> **Part 2**
>
> ### **(W2, Q1) Extra computation time of Cayley transformation**
>
> This particular effect observed by the reviewer is not caused by the Cayley transformation itself since the amount of computation is needed for any input batch size.
>
> We guess that the reviewer’s observation is related to some  software and/or GPU hardware features: modern ML frameworks and GPUs are highly optimized for particular matrix multiplication shapes, so the runtime does not always decrease monotonically with decreasing parameter dimension.  We also observed a similar phenomenon for the rank hyper-parameter. A similar pattern appears for the rank hyperparameter: as shown in Table 3 of the paper, at rank $r=8$, DoRA, PiSSA, and NB-LoRA all train faster than at neighboring ranks.
>
> In our experiments, we used micro-batch sizes of 4 (LLaMA-3-70B) and 8 (LLaMA-2-7B). As reported in Tables 1 and 3, NB-LoRA and LoRA have comparable overall computational cost. This matches the reviewer’s observation with microbatch size 8.
>
> ### **(Q2) LoRA and PiSSA on LLaMA-3-70B**
>
> PiSSA avoids the small-gradient issue at initialization, but for larger learning rates it produces large gradients which could lead to performance degradation, see our response to Reviewer SKyQ. In Table 1, we observe that PiSSA with lr=5e-5 still underperforms vanilla LoRA for LLaMA-3-70B. By further reducing lr to 2e-5 (the default in [1]), we can observe that PiSSA outperforms LoRA. The best GSM8K accuracy reported for PiSSA in [1] is 86.05\%, which is slightly lower than the best of vanilla LoRA (86.2\%) and the proposed NB-LoRA (87.1\%) at lr=5e-5.
>
> ### **(Q3) Difference of Figure 4a and Figure 3a**
>
> The two figures use different NB-LoRA norm types: Figure 3a (nuclear norm with $\delta=128$) and Figure 4a (Frobenius norm with $\delta=10$). Thus, their performance is slightly different.
>
> ### **(Q4) imbalance between $A$ and $B$**
>
> The gradients for LoRA has the form of
> $$
> \frac{\partial \ell}{\partial A}=\frac{\alpha}{r}B \left(\frac{\partial \ell}{\partial y}\right)x^\top,\quad \frac{\partial \ell}{\partial B}=\frac{\alpha}{r}Ax\left(\frac{\partial \ell}{\partial y}\right)^\top.
> $$
> Our motivating analysis shows that if $A$ is small and
> $B$ is initialized to zero, then both $\partial \ell /\partial A$ and $\partial \ell/\partial B$ are near zero, leading to slow optimization progress. With the Cayley transform, we cannot initialize both $A$ and $B$ to small values. Under the NBLoRA initialization, $B$ is zero but $A$ is orthogonal, and thus not small. Consequently, $\partial \ell/\partial B$ is relatively large at initialization, helping $B$ escape the zero point quickly. Meanwhile, the gradient on $A$ increases accordingly. Such behavior can be found in **Fig. 3 (col. 3 - 4) in the supplementary material**. More detail analysis of the training dynamics and their connections to the motivating intuition can be found our response to Reviewer SKyQ.
>
> - **Regarding the additional note**
>
> Thank you for pointing this out. The API key has now been invalidated and removed.

---

> > ### Comment · Reviewer_xuPe · 2025-11-22
> >
> > Thanks for the authors' detailed response. However, my concerns have not been fully resolved.
> >
> > ### **W1**
> > Although both LoRA and NB-LoRA involve two hyperparameters, their roles are fundamentally different. In NB-LoRA, $A, B$ are constrained and the value of $\delta$ explicitly bounds the maximum norm of $\Delta W$ (this is  also the motivation of NB-LoRA), leading to a much stronger form of control. As Reviewer _smpu_ pointed out, in some scenarios a larger update magnitude may be desirable, so choosing an appropriate $\delta$ is crucial in practice.
> >
> > The authors provide an ablation study to investigate the influence of different hyperparameters, and I appreciate this effort. However, because the optimization structures of LoRA and NB-LoRA differ, using the same learning rate does not imply that the magnitude of updates to $A, B$ is comparable across the two methods.  What is more interesting, in my view, is that the best performance of the LoRA baseline consistently appears at the largest $\alpha$. Since $\alpha$ also affects the effective update magnitude (in a way similar to the learning rate), this suggests that LoRA might be under-tuned in the current experimental setup.
> >
> > For this reason, I would encourage the authors to add further experiments to more convincingly validate the effectiveness of NB-LoRA. For example, on LLaMA-2-7B, one could explore different learning rates (e.g., 1e-4, 2e-4, 5e-4) and ranks (e.g., 8 and 128), combined with different choices of $\alpha$ or $\delta$, and compare the resulting performance of LoRA and NB-LoRA under better-tuned conditions.
> >
> > ### **W2, Q1**
> > Precisely because the amount of computation required by NB-LoRA does not depend on the input batch size, understanding why different micro-batch sizes lead to noticeably different per-step training time is important. This directly affects the practical usability of the method in real-world training pipelines. I would appreciate it if the authors could provide further analysis or optimizations to clarify whether the observed performance variations arise from implementation issues or from bottlenecks inherent to the GPU hardware. By the way, have the authors also observed a similar trend when using a micro batch size of 1?
> >
> > ### **Q2**
> > The authors should expand the range of learning rates and update Table 2 to provide a comprehensive comparison.

---

> > > ### Author Response · Authors · 2025-11-27
> > > **Part 1**
> > >
> > > Thank you for your follow-up comments and suggestions. Below we provide our detailed responses.
> > >
> > > ### **W1**
> > >
> > > Following the reviewer’s suggestion, we conducted additional experiments on LLaMA-2-7B with varying learning rates and ranks. Below we summarize our findings.
> > >
> > > - Rank $r=128$: LoRA with different $\alpha$ v.s. NB-LoRA with nuclear norm bound of $\delta=\alpha$
> > >
> > > |Hyper Param $\alpha$|16|32|64|128|256|
> > > |:-:|:-:|:-:|:-:|:-:|:-:|
> > > |LoRA (lr=5e-4)|54.7|57.9|58.3|58.3|**60.3**|
> > > |LoRA (lr=2e-4)|49.1|51.0|54.2|55.8|**58.6**|
> > > |LoRA (lr=1e-4)|46.8|48.7|52.2|54.0|**57.0**|
> > > |Hyper Param. $\delta$|16|32|64|128|256|
> > > |NB-LoRA (lr=5e-4)|55.8|58.2|**60.6**|59.7|58.8|
> > > |NB-LoRA (lr=2e-4)|51.4|54.5|58.1|59.1|**61.1**|
> > > |NB-LoRA (lr=1e-4)|51.4|54.7|57.8|**60.1**|59.4|
> > >
> > > **Discussion.** Results are consistent with our previous hyper-parameter tuning experiments. LoRA improves its performance with larger learning rates and scaling parameter $\alpha$. For each choice of learning rate, NB-LoRA outperforms LoRA in terms of best, worst and average GSM8K accuracies. Moreover, the hyper parameter $\delta$ behaves like a regularization coefficient for NB-LoRA.
> > >
> > > - Rank $r=8$: LoRA with different $\alpha$ v.s. NB-LoRA with **spectral norm** bound of $\delta=\alpha/r$.
> > >
> > > |Hyper Param $\alpha$|4|8|16|32|64|
> > > |:-:|:-:|:-:|:-:|:-:|:-:|
> > > |LoRA (lr=5e-4)|49.4|50.9|54.6|56.1|**56.6**|
> > > |LoRA (lr=2e-4)|43.1|44.2|49.1|50.3|**53.7**|
> > > |LoRA (lr=1e-4)|38.2|41.8|41.2|46.4|**48.2**|
> > > |Hyper Param. $\delta$|0.5|1|2|4|8|
> > > |NB-LoRA (lr=5e-4)|48.8|54.2|55.0|55.5|**56.1**|
> > > |NB-LoRA (lr=2e-4)|47.4|48.9|53.0|55.0|**55.6**|
> > > |NB-LoRA (lr=1e-4)|42.2|47.2|47.9|52.2|**54.0**|
> > >
> > > **Discussion.** With a small rank budget, spectral-norm NB-LoRA $W_{\mathrm{nblora}}=2\delta \hat{B}^\top \hat{A}$ has a much more constrained hypothesis class than unconstrained LoRA $W_{\mathrm{lora}}=\frac{\alpha}{r}B^\top A$ as all singular values of $2\hat{B}\hat{A}$ are constrained with $[0,1]$. Despite its restricted model expressivity, NB-LoRA matches or exceeds LoRA in most settings expect $\delta=0.5, 4, 8$ and lr=5e-4. This indicates that the Cayley transformation based reparameterization yields more effective exploration of the low-rank and norm-constrained set of weight adaptation.
> > >
> > > When the downstream task shares similar distribution as the pretraining dataset, then two recent influential works [1, 2] show that the **global minimum of fine-tuning has low rank and small magnitude** while spurious local minima (if they exist) have high rank and large magnitude. Under this regime, NB-LoRA's explicit norm control and optimization acceleration (see more details in the response to Reviewer SKyQ) is beneficial.
> > >
> > > However, as noted by Reviewer smpu, when the downstream data contains new knowledge that requires large singular values,  vanilla LoRA may indeed be more suitable than NB-LoRA. We will make this point clear in the limitation discussion of the revised version.
> > >
> > > **References:**
> > >
> > > [1] Jang et al. LoRA Training in the NTK Regime has No Spurious Local Minima. ICML 2024 (oral).
> > >
> > > [2] Kim et al. Lora training provably converges to a low-rank global minimum or it fails loudly (but it probably won’t fail). ICML 2025 (oral).

---

> > > > ### Author Response · Authors · 2025-11-27
> > > > **Part 2**
> > > >
> > > > ### **W2, Q1**
> > > >
> > > > We performed a comprehensive set of timing ablations with varying batch size, hardware setup, and implementation choice (custom backward vs auto-diff). Across all experiments, we observed the same qualitative trend reported by the reviewer:
> > > >
> > > > - **When the batch size is very small (1–2), there is a big runtime gap between NB-LoRA and LoRA; the gap narrows for larger batch sizes.**
> > > >
> > > > Beside the above observation, we also have two findings:
> > > >
> > > > 1. **This phenomenon is not unique to NB-LoRA** as it also occurs for other PEFT methods (e.g. DeLoRA, see Experiment 5);
> > > > 2. **Our custom backward pass can reduce NB-LoRA's overhead by ~40\%** compared with auto-diff.
> > > >
> > > > Our experiments start with modify the training parameters used in in Table 3 of the paper: 4$\times$H200, per device batch size (16), gradient accumulation steps (2), rank $r=128$, and training steps (781). All experiment tables are included above for completeness.
> > > >
> > > > - Experiment 1: gradient accumulation steps (2 $\rightarrow$ 1), training steps (781 $\rightarrow$ 100).  we report the overall train time (s) over 100 steps.
> > > >
> > > > |Batch size|1|2|3|4|5|6|7|8|
> > > > |:-:|:-:|:-:|:-:|:-:|:-:|:-:|:-:|:-:|
> > > > |LoRA|19.8|20.3|24.1|29.0|34.1|38.3|43.3|47.5|
> > > > |NB-LoRA|34.4|34.1|36.2|40.6|44.8|49.5|54.8|59.1|
> > > > |Difference|14.6|13.8|12.1|11.6|10.7|11.2|11.5|11.6|
> > > >
> > > > - Experiment 2: 4$\times$H200 $\rightarrow$ 1$\times$H200.
> > > >
> > > > |Batch size|1|2|3|4|5|6|7|8|
> > > > |:-:|:-:|:-:|:-:|:-:|:-:|:-:|:-:|:-:|
> > > > |LoRA|16.5|16.9|19.5|23.5|27.3|31.7|36.4|40.7|
> > > > |NB-LoRA|30.7|32.1|32.3|35.1|38.7|43.8|48.2|52.7|
> > > > |Difference|14.2|15.2|12.8|11.6|11.4|12.1|11.8|12.0|
> > > >
> > > >
> > > > - Experiment 3: 1$\times$H200 $\rightarrow$ 1$\times$A100 SXM4.
> > > >
> > > > |Batch size|1|2|3|4|5|6|7|8|
> > > > |:-:|:-:|:-:|:-:|:-:|:-:|:-:|:-:|:-:|
> > > > |LoRA|21.2|23.4|32.3|41.7|51.5|63.1|73.8|83.7|
> > > > |NB-LoRA|44.1|44.6|51.4|59.9|69.7|81.9|92.3|102.6|
> > > > |Difference|22.9|21.2|19.1|18.2|18.2|18.8|18.5|18.9|
> > > >
> > > > - Experiment 4: NB-LoRA (custom backward pass $\rightarrow$ automatic differentiation)
> > > >
> > > > |Batch size|1|2|3|4|5|6|7|8|
> > > > |:-:|:-:|:-:|:-:|:-:|:-:|:-:|:-:|:-:|
> > > > |LoRA|21.2|23.4|32.3|41.7|51.5|63.1|73.8|83.7|
> > > > |NB-LoRA (auto-diff)|58.5|59.5|65.5|73.9|83.9|95.0|105.7|115.6|
> > > > |Difference|37.3|36.1|33.2|32.2|32.4|31.9|31.9|31.9|
> > > >
> > > > - Experiment 5: NB-LoRA $\rightarrow$ DeLoRA. A similar trend as the previous experiments is observed. An additional interesting phenomenon is that, at a batch size of 8, DeLoRA outperforms LoRA by 1.4s-beyond the variance range. This may be due to low-level code optimization within the ML software framework or other factors outside the authors' current understanding.
> > > >
> > > > |Batch size|1|2|3|4|5|6|7|8|
> > > > |:-:|:-:|:-:|:-:|:-:|:-:|:-:|:-:|:-:|
> > > > |LoRA|21.2|23.4|32.3|41.7|51.5|63.1|73.8|83.7|
> > > > |DeLoRA|30.4|32.1|36.2|44.4|52.9|63.5|73.6|82.3|
> > > > |Difference|9.2|8.7|3.9|2.7|3.4|0.4|-0.2|-1.4|
> > > >
> > > > ### **Q2**
> > > >
> > > > Because LoRA often performs better under larger learning rates, we adjusted the learning rates in Table 2 to ensure a fair comparison: lr=2e-4 for Mistral and lr=7e-4 for LLaMA family. These were obtained by starting from lr = 5e-4 (Mistral) and lr = 1e-3 (LLaMA), then decreasing until LoRA training became stable. The updated results (included in the **supplementary material**) show that NB-LoRA still achieves the best average performance across tasks in most settings, though the performance gap narrows slightly.

---

> > > > > ### Comment · Reviewer_xuPe · 2025-11-27
> > > > >
> > > > > ### **W2, Q1**
> > > > > Considering such phenomenon is also observed for DeLoRA , I think maybe further discussion about training time is beyond the scope of this paper. I also appreciate the authors' work on the custom backward pass.
> > > > >
> > > > > ### **Q2**
> > > > > Sorry about the typo in the last comment. Could you please provide the results of Table **1**  with a larger learning rate range for a comprehensive comparison?

---

> > > > > > ### Author Response · Authors · 2025-12-04
> > > > > >
> > > > > > By following the reviewer's suggestion, we added the results with expanded learning rate range in the revised version.

---

> > > > ### Comment · Reviewer_xuPe · 2025-11-27
> > > >
> > > > As dicussed in  [1], $\alpha$  could also affect the effective update magnitude (in a way similar to the learning rate). According to the additional experiments, one can expect with larger $\alpha$, LoRA could have a better performance. This suggests that LoRA may still be under-tuned in the current setting.  To more convincingly demonstrate the fairness of the comparison, I would at least expect to see a U-shaped performance curve, where the performance of LoRA and NB-LoRA first increases and then decreases as the learning rate and/or $\alpha$ grow. Such a result would indicate that both methods are evaluated near their respective optima rather than in a potentially suboptimal regime. I look forward to seeing further experimental results along these lines.
> > > >
> > > > [1] The Primacy of Magnitude in Low-Rank Adaptation. NeurIPS 2025 spotlight.

---

> > > > > ### Author Response · Authors · 2025-12-04
> > > > >
> > > > > We thank the reviewer for  pointing out Ref. [1]. Its analysis is insightful and closely related to our work, and we will cite it in the revised version.
> > > > >
> > > > > Following the reviewer's suggestion, we conducted additional hyper-parameter searching experiments, including $\alpha=512, 1024$. As the reviewer anticipated, we observed a U-shape performance curve for LoRA, see the table below. We did not include the results for $\alpha=1024$ since some of them exhibited training instability.
> > > > >
> > > > > |Hyper Param. $\alpha$ |  32 | 64| 128 | 256| 512|
> > > > > |:-:|:-:|:-:|:-:|:-:|:-:|
> > > > > |LoRA (lr=5e-4)|57.9 | 58.3 | 58.3 | **60.3** | 57.8 |
> > > > > |LoRA (lr=2e-4) | 51.0 |54.2 | 55.8 | 58.6 | **59.9**|
> > > > > |LoRA (lr=1e-4)| 48.7 |52.2 | 54.0 |**57.0** | 56.6|
> > > > > |Hyper Param. $\delta$ |  16 | 32 | 64| 128 | 256|
> > > > > |NB-LoRA (lr=5e-4)  |55.8 | 58.2 | **60.6** | 59.7 | 58.8 |
> > > > > |NB-LoRA (lr=2e-4)  |51.4 | 54.5 | 58.1 | 59.1 | **61.1**|
> > > > > |NB-LoRA (lr=1e-4) |51.4 |54.7 |57.8 | **60.1**| 59.4|
> > > > >
> > > > > We also note that Table 2 in our submission uses different base models, yet the U-shaped behavior is already apparent under the current setup. Further increasing $\alpha$ tends to exacerbate instability rather than improve performance.

---

### Meta-Review · Area_Chair_Skts · 2025-12-19

**Summary:**

The paper introduces Norm-Bounded Low-Rank Adaptation (NB-LoRA), a novel parameterization for fine-tuning large models that enforces explicit bounds on the singular values of the adaptation matrix via the Cayley transform. The authors aim to address the sensitivity of standard LoRA to hyperparameters, such as learning rates, and to mitigate catastrophic forgetting by constraining the norm of the weight updates. The submission includes theoretical guarantees regarding the smoothness and completeness of the parameterization, alongside empirical evaluations on Large Language Models and Vision Transformers.


Having carefully read the original manuscript, the supplementary materials, and the extensive exchanges between the authors and the four reviewers at least three times, I have synthesized the assessment. The reviewers presented a divided perspective. Reviewer HAkH championed the work for its elegant theoretical basis and the practical utility of the custom backward pass, eventually raising their score to an acceptance level after the authors included low-data regime experiments. Conversely, Reviewers xuPe, SKyQ, and smpu maintained reservations regarding the necessity of the method, the fairness of the baseline comparisons, and the fundamental trade-off between norm constraints and model expressivity.

**Reviewer Concerns:**

The rebuttal phase was active, with the authors conducting additional experiments to address specific queries, yet several critical concerns remain outstanding.

Reviewer HAkH’s concern regarding the lack of evaluation in low-data regimes was successfully addressed by the authors, who added DreamBooth experiments demonstrating that NB-LoRA can prevent overfitting better than baselines. This effectively resolved HAkH's primary hesitation.

However, the concerns raised by Reviewers xuPe and smpu regarding the experimental baselines and the conceptual necessity of the method were not fully resolved. Reviewer xuPe pointed out that standard LoRA appeared under-tuned in the main comparisons; specifically, that LoRA's performance improves with a higher scaling factor ($\alpha$), suggesting that the reported gaps between NB-LoRA and LoRA might be artifacts of suboptimal baseline hyperparameters. While the authors provided additional ablation studies showing a U-shaped performance curve, this confirmed that the original comparisons did not capture LoRA's peak potential, weakening the empirical claims of superiority.

Furthermore, Reviewer smpu raised a fundamental theoretical objection that remains the primary stumbling block for acceptance: the necessity and desirability of strict norm bounds. The reviewer argued that in many fine-tuning scenarios, particularly those requiring the injection of new knowledge or adaptation to significantly different distributions, large spectral updates are necessary. Constraining the norm might hinder the model's expressivity. The authors conceded this limitation in their response, acknowledging that for tasks requiring high-spectral-norm updates, vanilla LoRA might be superior. This concession, while honest, narrows the scope of the paper's contribution significantly. Additionally, the argument that NB-LoRA improves robustness to learning rates was met with the counter-argument that learning rate sensitivity can be managed via schedulers or tuning, without incurring the computational overhead and implementation complexity of the Cayley transform.

Finally, while Reviewer SKyQ appreciated the improved analysis of training dynamics (shifting from raw gradients to weight norms), their score remained unchanged, indicating that while the analysis was corrected, it did not sufficiently strengthen the core value proposition of the paper to warrant acceptance.

**Reviewer Scores:**

Reviewer HAkH (Score: 8): This reviewer was satisfied with the rebuttal, particularly the addition of low-data experiments, and would likely advocate for acceptance.

Reviewer xuPe (Score: 4): This reviewer would likely maintain their score, as their concerns about the fairness of the baseline tuning and the computational overhead (which persists despite explanations) were not fully alleviated.

Reviewer SKyQ (Score: 4): This reviewer would likely maintain their score, finding the method technically sound but lacking in a compelling causal link between the proposed mechanism and the claimed benefits that outweighs the complexity.

Reviewer smpu (Score: 2): This reviewer would likely maintain a rejection stance, remaining unconvinced that the problem NB-LoRA solves (norm control) is a problem that needs solving via complex parameterization rather than simple hyperparameter tuning.

---

### Decision · Program_Chairs · 2026-01-26

Reject